# SOX9-dependent fibrosis drives renal function in nephronophthisis

Maulin Mukeshchandra Patel[1], Vasileios Gerakopoulos[1], Bryan Lettenmaier[1], Eleni Petsouki[1], Kurt A Zimmerman[2], John A Sayer [ID][3,4,5] & Leonidas Tsiokas [ID][1✉]

## Abstract

Fibrosis is a key feature of a broad spectrum of cystic kidney diseases, especially autosomal recessive kidney disorders such as nephronophthisis (NPHP). However, its contribution to kidney function decline and the underlying molecular mechanism(s) remains unclear. Here, we show that kidney-specific deletion of *Fbxw7*, the recognition receptor of the SCF$^{FBW7}$ E3 ubiquitin ligase, results in a juvenile-adult NPHP-like pathology characterized by slow-progressing corticomedullary cysts, tubular degeneration, severe fibrosis, and gradual loss of kidney function. Expression levels of SOX9, a known substrate of FBW7, and WNT4, a potent pro-fibrotic factor and downstream effector of SOX9, were elevated upon loss of FBW7. Heterozygous deletion of *Sox9* in compound mutant mice led to the normalization of WNT4 levels, reduced fibrosis, and preservation of kidney function without significant effects on cystic dilatation and tubular degeneration. These data suggest that FBW7-SOX9-WNT4-induced fibrosis drives kidney function decline in NPHP and, possibly, other forms of autosomal recessive kidney disorders.

**Keywords** FBW7; SOX9; Fibrosis; Nephronophthisis; Kidney Function
**Subject Category** Urogenital System

## Introduction

Renal fibrosis is a key pathological feature and a critical predictor of disease progression and poor patient outcomes (Attanasio et al, 2007; Fragiadaki et al, 2020; Gupta et al, 2021; Li et al, 2021; Wen, 2011; Wolf et al, 2024; Xue and Mei, 2019) across various cystic kidney diseases, especially the autosomal recessive forms such as nephronophthisis (NPHP) (Attanasio et al, 2007; Gupta et al, 2021; Li et al, 2021) and autosomal recessive polycystic kidney disease (ARPKD) (Wen, 2011). NPHP is particularly distinguished by its pronounced fibrotic response relative to cystic expansion (Slaats

et al, 2016; Srivastava et al, 2017). NPHP affects ~1 in 50,000 children and young adults and is frequently associated with syndromic forms of Polycystic Kidney Disease (PKD) like Joubert, Bardet-Biedl, and Meckel-Gruber syndromes (Attanasio et al, 2007; Gana et al, 2022; Li et al, 2021; Li et al, 2023; Srivastava et al, 2017; Van De Weghe et al, 2022; Wolf and Hildebrandt, 2011; Wolf et al, 2024). Although the severity of fibrosis and cystic expansion differs among these conditions, there is a consistent and significant correlation between the extent of fibrosis and the decline in kidney function. Interestingly, the causal role of fibrosis in kidney function decline in these diseases has not yet been examined.

NPHP is characterized by slow-progressing corticomedullary cysts, tubular degeneration, extensive tubulointerstitial fibrosis, and a gradual decline in kidney function (Petzold et al, 2023; Salomon et al, 2009; Stokman et al, 1993; Van De Weghe et al, 2022; Wolf and Hildebrandt, 2011). Mutations in more than 20 genes (Srivastava et al, 2017; Wolf et al, 2024) have been associated with NPHP, which are primarily shown to cause ciliary dysfunction. Although these studies have shed light on the root cause of the disease, they have not yet clearly identified the exact molecular mechanisms downstream of ciliary dysfunction that trigger cyst formation, fibrosis, or tubular damage, nor have they determined the impact of these pathologies on kidney function. This knowledge gap is mostly due to a lack of knowledge of the molecular function of some of these genes or, for the ones whose molecular function is known, inconclusive information about their cellular roles downstream of cilia (Attanasio et al, 2007; Jiang et al, 2008; Kishimoto et al, 2017; Konig et al, 2017; Li et al, 2021; Louie et al, 2010). For example, while we know the molecular functions of IFT74 (JBST40) (Boegholm et al, 2023; Luo et al, 2021) or CEP290 (NPHP6) (Barbelanne et al, 2015; Helou et al, 2007; Prosser et al, 2022; Tsang et al, 2008) in cilia, this knowledge is not sufficient enough to help us decipher the cellular pathway(s) that lead from defective cilia to cystogenesis, fibrosis, and/or tubular degeneration. In addition, NPHP1 orthologous mouse models fail to accurately mimic the early kidney function decline seen in patients (Jiang et al, 2008; Li et al, 2021). Therefore, innovative approaches are needed to identify the relevant cellular pathways that connect organelle to organ biology, and disentangle the roles of cystogenesis, fibrosis, and degeneration in kidney function.

[1]Department of Cell Biology, University of Oklahoma Health Sciences Center, Oklahoma City, OK, USA. [2]Department of Internal Medicine, Division of Nephrology, University of Oklahoma Health Sciences Center, Oklahoma City, OK, USA. [3]Biosciences Institute, Faculty of Medical Sciences, Newcastle University, Newcastle upon Tyne, UK. [4]Renal Services, Newcastle Upon Tyne Hospitals NHS Foundation Trust, Newcastle upon Tyne, UK. [5]NIHR Newcastle Biomedical Research Centre, Newcastle upon Tyne, UK.
✉E-mail: leonidas-tsiokas@ouhsc.edu

FBW7 (F-box and WD repeat domain-containing 7) is the recognition receptor of the SCF (SKP1-CUL1-F-box) E3 ligase complex (SCF$^{FBW7}$). It recognizes target proteins through a conserved and well-characterized phosphodegron motif. By controlling protein levels of numerous targets, FBW7 contributes to cellular proteostasis regulating many basic cell functions such as proliferation, metabolism, differentiation, and cell death in a cell context-dependent manner (Davis et al, 2014; Shimizu et al, 2018; Wang et al, 2011). We have shown that FBW7 functions as a rheostat for ciliary length, influencing ciliary structure and signaling output (Maskey et al, 2015; Petsouki et al, 2021). Because of the wide range of proteins processed by FBW7 and its known molecular function as a component of an E3 ubiquitin ligase, we reasoned that if *Fbxw7* deletion in the kidney results in cystic disease, this model could serve as an unbiased discovery platform to identify protein networks relevant to the pathogenesis of cystic kidney diseases.

Our study shows that deletion of *Fbxw7* in mouse kidney epithelial cells results in a juvenile–adult NPHP-like pathology recapitulating the typical triad of NPHP symptoms such as slow-progressing corticomedullary cysts, tubular degeneration, excessive interstitial fibrosis, and most importantly an early decline in kidney function. Guided by changes in steady-state expression levels of known FBW7 substrates in cystic tubules, we identified the SOX9-WNT4 axis as a crucial regulator of tubulointerstitial fibrosis and showed that fibrosis is the primary driver of the kidney function decline in NPHP. Importantly, we also observed elevated SOX9 expression in NPHP patient biopsy and a cilia-mutant mouse model, supporting the clinical relevance and broad impact of SOX9 on NPHP-like diseases.

# Results

## Deletion of *Fbxw7* in the mouse kidney leads to slow-progressing cystic disease and loss of kidney function

We deleted the *Fbxw7* gene in kidney epithelial cells using *Cdh16Cre*, which is constitutively active from E10.5 (Shao et al, 2002). This driver is active in epithelial cells of developing nephrons, the ureteric bud, and mesonephric tubules with low recombination efficiency in the proximal tubules. In the adult kidney, this promoter is active in the collecting ducts, loops of Henle, distal tubules, and to a lesser degree, in proximal tubules (Shao et al, 2002) (Appendix Fig. S1A–D). *Cdh16Cre;Fbxw7$^{f/f}$* mice displayed slow-progressing corticomedullary cysts with a gradual decline in kidney function. No changes were observed in the two kidneys-to-body weight ratio (2KW/BW) compared to *Fbxw7$^{f/f}$* mice (Fig. 1A–E). Tubular dilatations or mild cysts, particularly in the corticomedullary area, were evident as early as 3 months, whereas obvious cystic tubules were observed at 7 and 10 months (Figs. 1A and EV1A). Cysts were present in LTA-positive proximal tubules, NKCC2-positive loop of Henle, and DBA-positive collecting ducts (Fig. EV1B–J). Kidney function started to deteriorate as early as 3 months of age and continued to decline by 7 and 10 months of age, as represented by elevated levels of serum BUN, Creatinine, and Cystatin C (Fig. 1C–E). Glomerular cysts and Bowman's capsule thickening also supported reduced kidney function (Fig. EV2). Heterozygous *Cdh16Cre;Fbxw7$^{+/f}$* mice exhibited no significant changes in cyst development, 2KW/BW, or kidney function at 3 and 7 months of age (Appendix Fig. S2).

Next, we examined cyst formation and cell proliferation using *Cdh16Cre;Rosa$^{mT/mG}$;Fbxw7$^{f/f}$* mice, allowing us to trace the tubules that were affected by *Cdh16Cre*-induced recombination. From 3 months onward, cyst formation in GFP-positive tubules was accompanied by an upsurge in Ki67-positive cells. A modest increase in GFP-negative cells was also observed. These data suggested an increased proliferation in the *Cdh16Cre;Rosa$^{mT/mG}$;Fbxw7$^{f/f}$* mice (Fig. 1F,G; Appendix Fig. S3). This heightened proliferation was further supported by 5-ethynyl-2′-deoxyuridine (EdU) and phosphorylated histone H3 (phH3) staining to mark S-phase DNA synthesis and mitosis, respectively (Appendix Fig. S3A–E). These findings are consistent with the juvenile–adult form of NPHP characterized by slow-progressing cysts, increased proliferation, and a decline in kidney function.

## Deletion of *Fbxw7* causes increased tubular cell death, thickening of the tubular basal membrane, and urine defects

We noted that while *Cdh16Cre;Fbxw7$^{f/f}$* mice developed cysts and exhibited enhanced proliferation, there was no increase in 2KW/BW at 3, 7, or 10 months (Fig. 1B). This led us to investigate the possibility of increased cell death and tubular deterioration in these mice. TUNEL staining of 3 and 7 months *Cdh16Cre;Rosa$^{mT/mG}$;Fbxw7$^{f/f}$* kidneys revealed augmented cell death (Fig. 2A–D). Periodic acid-Schiff (PAS) staining showed disorganized architecture with tubular atrophy. Atrophic tubules had a narrow diameter, shrunken epithelium, or thickened/wrinkled basal membranes (Fig. 2E–H). We further examined urine parameters by collecting samples over 16 h using metabolic cages. *Cdh16Cre;Fbxw7$^{f/f}$* mice began showing urine defects at 3 months, which worsened at 7 and 10 months (Fig. 3). A significant reduction in urine specific gravity was observed in these mice, with no changes in the total protein-to-creatinine ratio, especially in aged mice (Fig. 3A–F). This defect was primarily driven by an increase in urine output, which correlated with elevated water consumption (Fig. 3G–L), suggestive of polyuria and polydipsia. The acidic urine pH (Fig. 3M–O) and low serum tCO$_2$ levels in aged mice (Appendix Fig. S4) further suggested acidosis resulting from altered renal handling of electrolytes and solutes. These pathological changes align with those seen in NPHP patients, where disrupted tubular function is often associated with polyuria and polydipsia. Tubular degeneration, structural changes in the tubular basement membrane, polyuria, and polydipsia are typical features of NPHP (Akira et al, 2021; Ala-Mello et al, 1996; Hammi et al, 2023; Shaik et al, 2020; Willemarck et al, 2010; Wolf and Hildebrandt, 2011).

## Development of inflammation and tubulointerstitial fibrosis upon deletion of *Fbxw7*

*Cdh16Cre;Rosa$^{mT/mG}$;Fbxw7$^{f/f}$* kidneys displayed a marked increase in both inflammation and fibrosis, particularly evident at the corticomedullary junction from 3 months of age. Specifically, *Cdh16Cre;Rosa$^{mT/mG}$;Fbxw7$^{f/f}$* kidneys exhibited a significant surge in F4/80 positive macrophages representing inflammation (Fig. 4A,B). Concurrently, there was a significant increase in Masson's trichrome staining, depicting collagen accumulation (Fig. 4C,D). In addition, we detected a considerable increase in Collagen-TypeI-Alpha1 (COL1A1) and α-SMOOTH MUSCLE ACTIN (α-SMA) expression, demonstrating an escalated fibrotic response upon loss of FBW7, especially in the

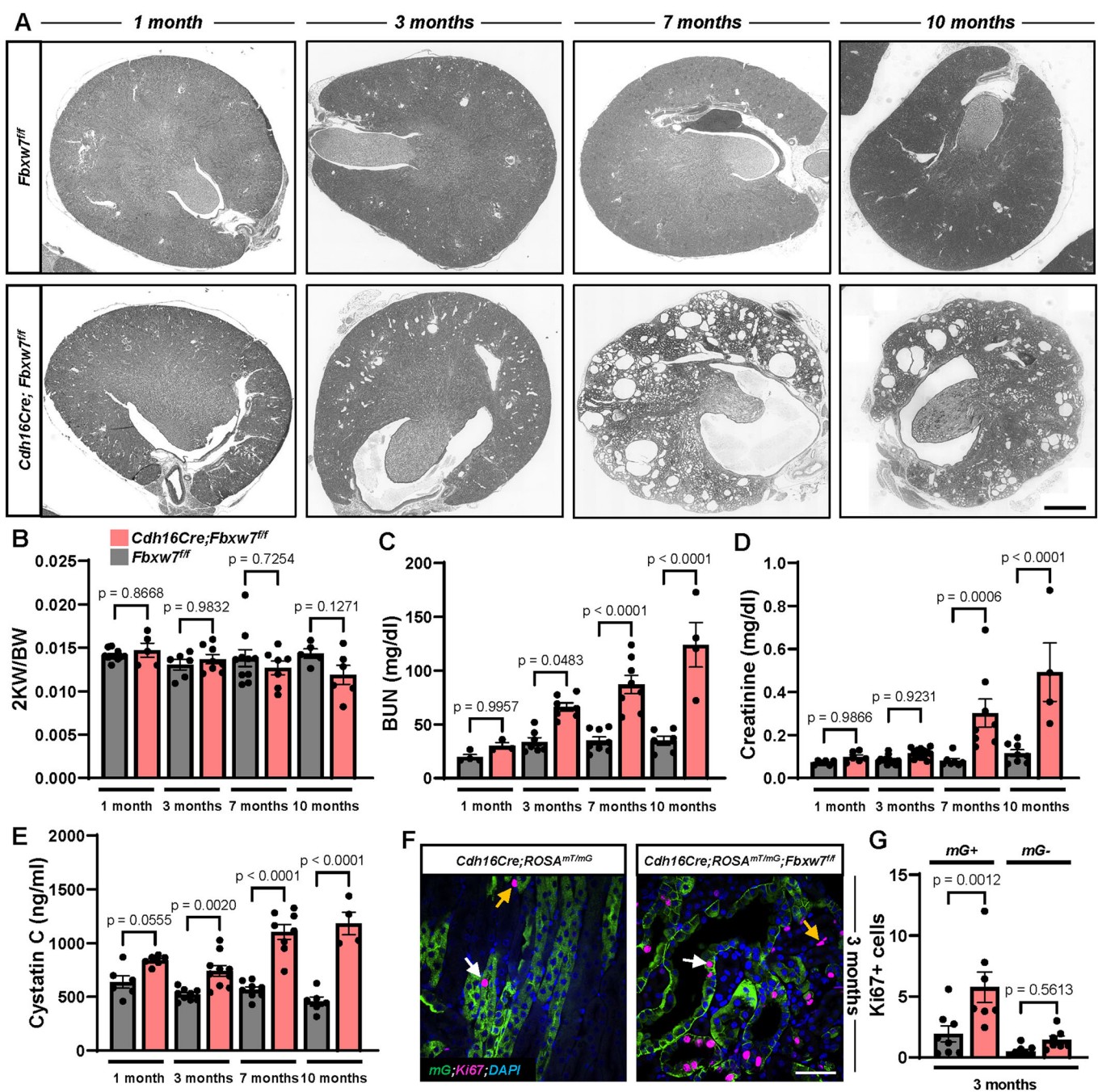

**Figure 1. Deletion of *Fbxw7* results in a slow-progressing cyst and a decline in kidney function.**

(A) Representative images of whole kidney section scan showing cyst progression, (B) 2 kidney weight/body weight (2KW/BW) ($n \geq 5$) (C) Serum BUN ($n \geq 3$), (D) Creatinine ($n \geq 4$), and (E) Cytastin C ($n \geq 4$) from 1-, 3-, 7-, and 10-month-old *Fbxw7^{l/l}* and *Cdh16Cre;Fbxw7^{l/l}* mice. Each data point represents one animal. Statistical analysis was performed using two-way ANOVA and is presented as the mean ± SEM. Scale bar: 400 µm. (F, G) Representative images and quantification of Ki67 staining from 3-month-old kidneys of *Cdh16Cre;ROSA^{mT/mG}* and *Cdh16Cre;ROSA^{mT/mG};Fbxw7^{l/l}* mice. Epithelial cells where *Cdh16Cre* is active express membrane-targeted GFP. (F) White arrows show Ki67-positive (+) (pink) epithelial cells where *Cdh16Cre* is active (*mG +*, green). Yellow arrows show Ki67+ cells where *Cdh16Cre* is inactive (*mG-*). Nuclei are stained with DAPI (blue). Scale bar: 50 µm. (G) Each data point represents the mean Ki67 +;mG+ or Ki67 +;mG- cells scored per animal ($n \geq 7$). Statistical analysis was performed using one-way ANOVA followed by Šídák's multiple comparisons test and is presented as the mean ± SEM. Source data are available online for this figure.

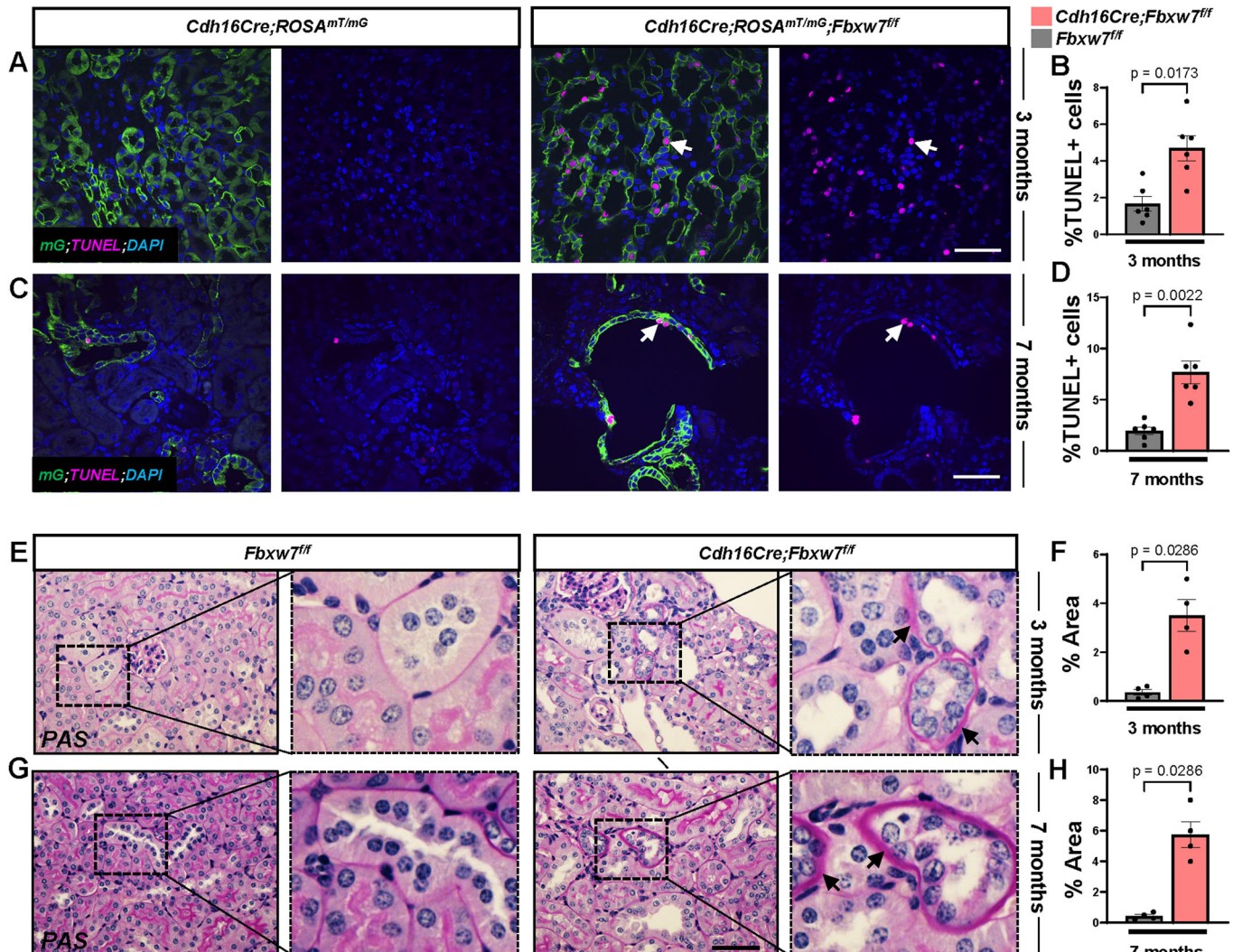

**Figure 2. Loss of FBW7 results in increased cell death and thickening of the tubular basal membrane.**

(A–D) Representative images of TUNEL staining and quantification from 3- and 7-month-old kidneys of *Cdh16Cre;ROSA^mT/mG* and *Cdh16Cre;ROSA^mT/mG;Fbxw7^f/f* mice. (A, C) White arrows show TUNEL+ (pink) cells in mG+ tubules (green). Nuclei are stained with DAPI (blue). Scale bar: 50 μm. (B, D) Each data point represents the percent of TUNEL+ cells from mG+ tubules scored per animal ($n \geq 6$). Statistical analysis was performed using the Mann–Whitney test and is presented as the mean ± SEM. (E–H) Representative images and quantification of tubules with thickened basement membrane using PAS staining from 3- and 7-month-old kidneys of *Fbxw7^f/f* and *Cdh16Cre;Fbxw7^f/f* mice. Black arrows show thickened tubular basement membranes (pink). A high-magnification image of the insets is shown on the right-side panel for each genotype. Nuclei are stained with Hematoxylin (blue). Scale bar: 200 μm. (F, H) Each data point represents a percentage of the area affected by tubules with thickened or wrinkled basement membranes scored per animal ($n \geq 4$). Statistical analysis was performed using the Mann–Whitney test and is presented as the mean ± SEM. Source data are available online for this figure.

corticomedullary and cortex region (Fig. 4E–H). This fibrotic response worsened at 7 months of age (Appendix Fig. S5). Overall, we established that loss of FBW7 mimics NPHP pathology comprising slow-progressing cysts, tubular degeneration, inflammation, and excessive fibrosis, all coupled with a gradual decline in kidney function.

## *Fbxw7* deletion results in the downregulation of TMEM237 and ciliary defects

To understand how loss of FBW7 results in NPHP pathology, we employed a quantitative proteomic approach, where protein expression profiles were compared between wild-type (parental mIMCD3 or

stable pools transfected with empty vector-mock) and *Fbxw7*-null (mIMCD3^Fbxw7-KO#2 and mIMCD3^Fbxw7-KO#7 clones) mIMCD3 cell lines (Appendix Fig. S6; Dataset EV1). Comprehensive pathway analysis revealed an upregulation of pathways like fatty acid oxidation and oxidative stress response alongside a downregulation of pathways associated with protein degradation machinery and DNA damage response, aligning with the established functions of FBW7 (Appendix Fig. S7) (Davis et al, 2018; Lan and Sun, 2021; Onoyama et al, 2011; Shimizu et al, 2018; Wei et al, 2023). Analysis of the top 20 up- or downregulated proteins pointed toward TMEM237 downregulation (Fig. 5A). Loss-of-function mutations in the *TMEM237* gene are prevalent in individuals with Joubert Syndrome, a condition known to

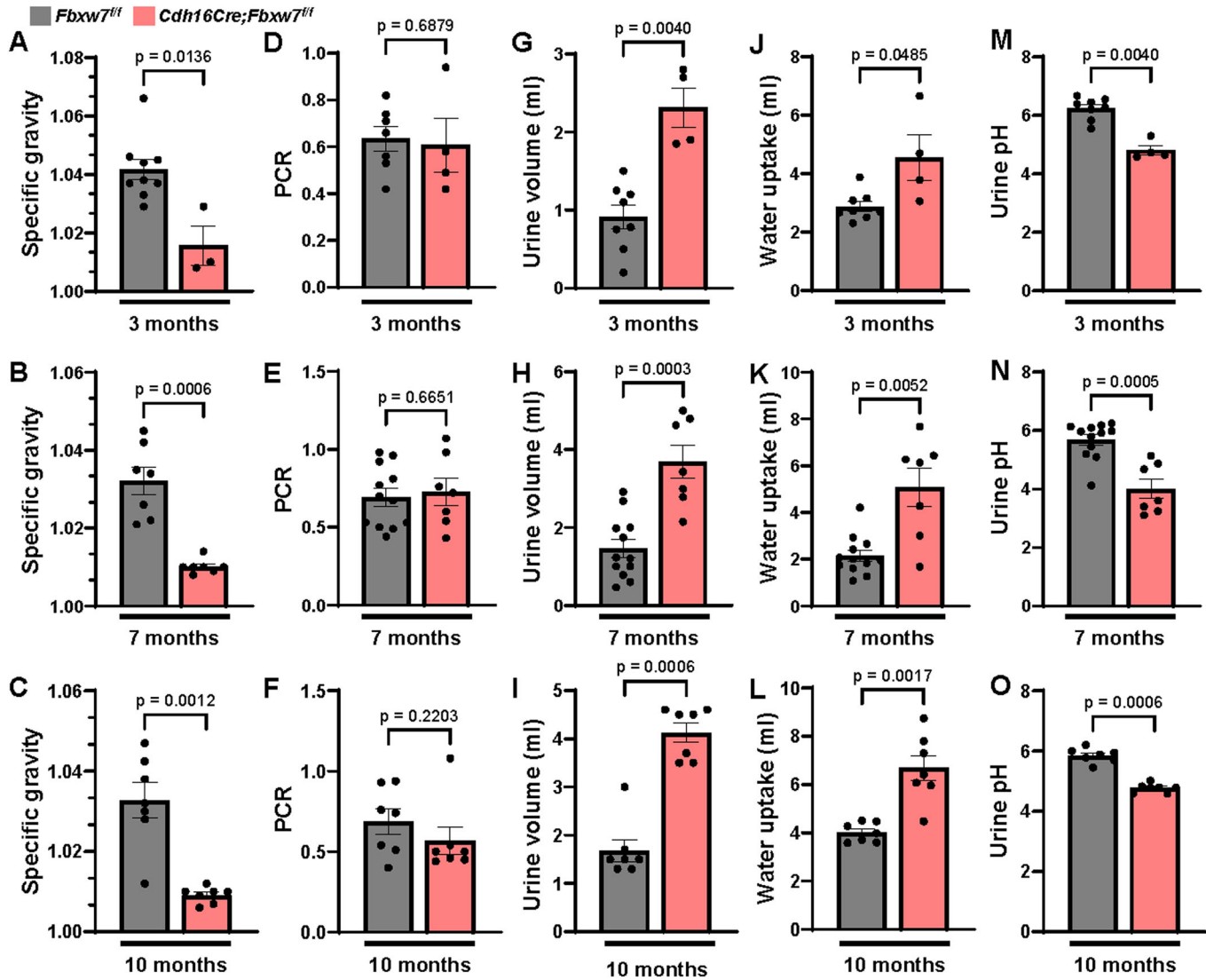

**Figure 3. Deletion of *Fbxw7* results in urine defects.**

Urine defects characterized by variations in (**A–C**) specific gravity ($n \geq 3$), (**D–F**) total protein-to-creatinine ratio (PCR) in urine ($n \geq 4$), (**G–I**) urine volume ($n \geq 4$), (**J–L**) water uptake ($n \geq 4$), and (**M–O**) urine pH ($n \geq 4$) from 3-, 7-, and 10-month-old *Fbxw7^{f/f}* and *Cdh16Cre;Fbxw7^{f/f}* mice using metabolic cages. Each data point represents one animal. Statistical analysis was performed using the Mann–Whitney test and is presented as the mean ± SEM. Source data are available online for this figure.

present with NPHP (Fleming et al, 2017; Gana et al, 2022; Huang et al, 2011; Van De Weghe et al, 2022). The downregulation of TMEM237 revealed by mass spectrometry was validated using immunoblotting of lysates of wild-type and *Fbxw7*-null mIMCD3 cells (Fig. 5B,C). Expression of TMEM237 was also drastically reduced in cystic tubules in the corticomedullary region in 3-month-old animals (Fig. 5D,E). Since patients with Joubert syndrome display a renal NPHP phenotype, these data provided mechanistic support, at least in part, for the NPHP-like phenotype induced by the deletion of *Fbxw7*. Since the loss of most NPHP-causing genes, including *Tmem237*, is shown to cause defective cilia (Huang et al, 2011), we further investigated if cilia were affected by the deletion of *Fbxw7*. We found that the cystic tubules in *Cdh16Cre;Rosa^{mT/mG};Fbxw7^{f/f}* kidneys (Fig. 5F–H), as well as *Fbxw7*-null mIMCD3 cells (Fig. 5I–K), displayed shorter cilia and reduced ciliation. Collectively, these findings suggest that the loss of

FBW7 mimics NPHP not only phenotypically but also mechanistically, including ciliary defects and downregulation of TMEM237, a bona fide ciliopathy gene.

## Deletion of *Fbxw7* induces SOX9-dependent fibrosis and kidney function decline

FBW7 functions as a substrate recognition component for the SCF^{FBW7} ubiquitin E3 ligase, and its absence typically leads to the accumulation of its target proteins. Leveraging this knowledge, we further investigated direct substrates of FBW7 whose accumulation could elucidate the various pathologies observed in our NPHP-like mouse model. SOX9 and c-MYC emerged as key proteins of interest due to several factors: both are well-established direct targets of FBW7 (Suryo Rahmanto et al, 2016; Welcker et al, 2004),

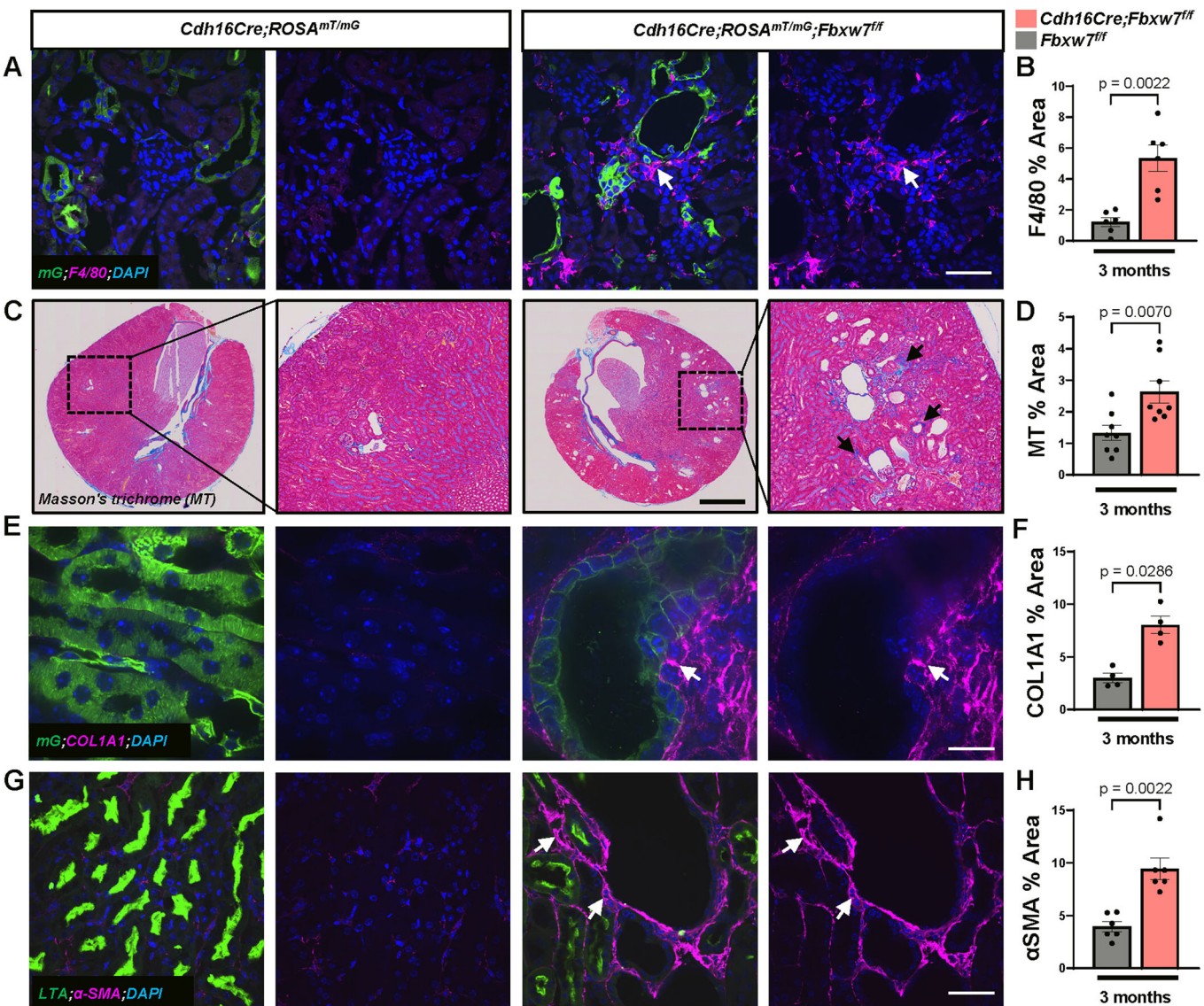

**Figure 4.** Deletion of *Fbxw7* causes inflammation and tubulointerstitial fibrosis.

(A, B) Representative images and quantification of F4/80 staining from 3-month-old kidneys of *Cdh16Cre;ROSA^{mT/mG}* and *Cdh16Cre;ROSA^{mT/mG};Fbxw7^{f/f}* mice. (A) White arrows show F4/80+ (pink) cells localized adjacent to the mG+ tubules (green). Nuclei are stained with DAPI (blue). Scale bar: 50 µm. (B) Each data point represents the percent corticomedullary and cortex area of F4/80+ cells per animal (*n* = 6). Statistical analysis was performed using the Mann–Whitney test and is presented as the mean ± SEM. (C, D) Representative images and quantification of Masson's trichrome (MT) staining from 3-month-old kidneys of *Fbxw7^{f/f}* and *Cdh16Cre;Fbxw7^{f/f}* mice. (C) the left side image shows the whole kidney section scan, and the right side image shows a high-magnification image of the insets. Black arrows show MT staining (purple). Scale bar: 400 µm. (D) Each data point represents the percent area of MT staining per whole kidney section scan per animal (*n* = 8). Statistical analysis was performed using the Mann–Whitney test and is presented as the mean ± SEM. (E, F) Representative images and quantification of COL1A1 staining from 3-month-old kidneys of *Cdh16Cre;ROSA^{mT/mG}* and *Cdh16Cre;ROSA^{mT/mG};Fbxw7^{f/f}* mice. White arrows show COL1A1 staining (pink) adjacent to the mG+ tubules (green). Nuclei are stained with DAPI (blue). Scale bar: 20 µm. (F) Each data point represents the COL1A1+ percent area per animal (*n* = 4). Statistical analysis was performed using the Mann–Whitney test and is presented as the mean ± SEM. (G, H) Representative images and quantification of α-SMA staining from 3-month-old kidneys of *Fbxw7^{f/f}* and *Cdh16Cre;Fbxw7^{f/f}* mice. White arrows show α-SMA staining (pink) adjacent to the LTA+ cortical region (green). Nuclei are stained with DAPI (blue). Scale bar: 50 µm. (H) Each data point represents the α-SMA+ percent area per animal (*n* = 6). Statistical analysis was performed using the Mann–Whitney test and is presented as the mean ± SEM. Source data are available online for this figure.

act as a mediator of fibrosis or cystogenesis (Aggarwal et al, 2024; Kumar et al, 2015; Parrot et al, 2019; Raza et al, 2021), and have not been investigated in the context of NPHP.

Given our observations of increased fibrosis and cysts in the corticomedullary region, we specifically investigated SOX9 and c-MYC expression in these areas. We found that c-MYC was

significantly increased in *Fbxw7*-null cystic tubules as well as in IMCD3 cells (Appendix Fig. S8A,B). To determine whether the increased expression of c-MYC contributes to NPHP pathology following *Fbxw7* deletion, we generated *Cdh16Cre;Fbxw7^{f/f};c-Myc^{+/f}* animals. No significant changes were observed in the cystic index, 2KW/BW, serum BUN, and Masson's trichrome staining

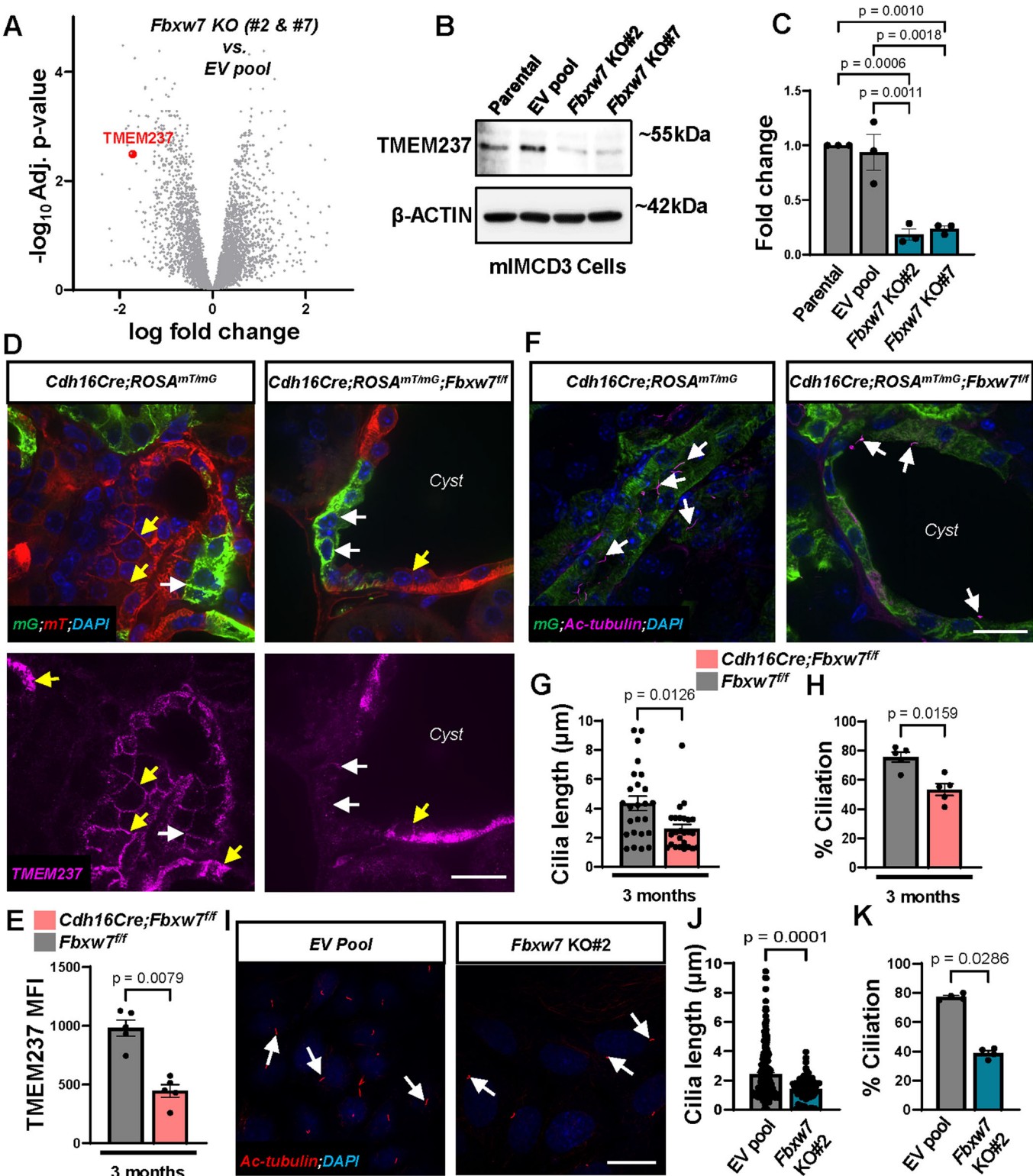

(Appendix Fig. S8C–G) in *Cdh16Cre;Fbxw7^{f/f};c-Myc^{+/f}* animals compared to *Cdh16Cre;Fbxw7^{f/f}* animals, suggesting that normalizing c-MYC levels do not significantly ameliorate early decline in kidney function and NPHP-like pathology seen in our mouse model.

Next, we examined SOX9 expression and observed a significant increase in both cystic and non-cystic *Fbxw7*-null kidney tubules in the corticomedullary and cortex regions (Figs. 6A,B and EV3). Additionally, we observed increased SOX9 expression in proximal tubules with brush borders and in ATP1A1-positive tubules that

**Figure 5. Loss of FBW7 results in TMEM237 downregulation coupled with ciliary defects.**

(A) Volcano plot showing differentially expressed proteins from *Fbxw7*-null cells (*Fbxw7 KO#2* and *Fbxw7 KO#7*) versus empty vector (EV) pool mIMCD3 cells. The red dot represents the TMEM237 protein that is significantly downregulated in *Fbxw7*-null cells compared to EV pool mIMCD3 cells. $n = 3$ independent experiments. The data was normalized using cyclic loess, and statistical analysis was performed using linear models for microarray data (limma) with empirical Bayes (eBayes) smoothing to the standard errors. Proteins with an FDR-adjusted *P* value <0.05 and a fold change >2 were considered significant. (B) Validation of quantitative proteomics using immunoblotting and (C) quantification of TMEM237 from $n = 3$ independent experiments. Statistical analysis was performed using one-way ANOVA followed by Šídák's multiple comparisons test and is presented as the mean ± SEM. (D, E) Representative images and quantification of TMEM237 staining from 3-month-old kidneys of *Cdh16Cre;ROSA^{mT/mG}* and *Cdh16Cre;ROSA^{mT/mG};Fbxw7^{f/f}* mice. (D) The white and yellow arrows show TMEM237 expression (pink) in cells where *Chd16Cre* is active (mG +, green) or inactive (mT +, red) within the same kidney tubule, respectively. Nuclei are stained with DAPI (blue). Scale bar: 20 μm. (E) Each data point represents an average of TMEM237 MFI per animal ($n = 5$) in cystic tubules that displayed partial *Cdh16Cre*-based recombination and were positive for TMEM237 staining. Statistical analysis was performed using the Mann–Whitney test and is presented as the mean ± SEM. (F–K) Representative images and quantification of acetylated-TUBULIN (cilia) staining from (F) 3-month-old kidneys of *Cdh16Cre;ROSA^{mT/mG}* and *Cdh16Cre;ROSA^{mT/mG};Fbxw7^{f/f}* mice and (I) *Fbxw7*-null mIMCD3 cells. (F) White arrows show cilia (pink) in mG+ cystic kidney tubules (green). Nuclei are stained with DAPI (blue). Scale bar: 10 μm. (I) White arrows show cilia (red) in mIMCD3 cells after 48 h of serum starvation. Nuclei are stained with DAPI (blue). Scale bar: 10 μm. (G, H) Each data point represents the mean ciliary length per field of view from >20 images or percent ciliation in cystic cells from $n \geq 3$ animals. Statistical analysis was performed using the Mann–Whitney test and is presented as the mean ± SEM. (J, K) Each data point represents the mean ciliary length from >70 cells or percent ciliation from >200 cells from $n \geq 3$ experiments. Statistical analysis was performed using the Mann–Whitney test and is presented as the mean ± SEM. Source data are available online for this figure.

had lost apicobasal polarity (Fig. EV3). The spatiotemporal analysis further revealed a gradual increase in SOX9-positive cells with aging, which strongly correlated with the increase in αSMA response seen in our mouse model (Fig. EV3; Appendix Fig. S9). These findings suggest that at least a subset of the observed SOX9-positive cells is unable to or has delayed epithelium repair and are actively contributing to fibrosis (Aggarwal et al, 2024). *Fbxw7*-null IMCD3 cells also displayed increased SOX9 levels (Fig. 6C,D). To determine whether the increased SOX9 contributes to NPHP pathology following *Fbxw7* deletion, we generated *Cdh16Cre;Fbxw7^{f/f};Sox9^{+/f}* animals. These animals showed amelioration of fibrosis at 3 months of age, confirmed by Masson's trichrome staining (Fig. 6E,F). Reduced fibrosis was also evidenced by decreased COL1A1 (Fig. 6G,H) and α-SMA (Fig. 6I,J) levels. Since SOX9 is shown to induce fibrosis by upregulating WNT4 (Aggarwal et al, 2024), we analyzed WNT4 expression. We found that the loss of FBW7 led to elevated WNT4 levels, which were subsequently ameliorated by the loss of SOX9 (Fig. 6K,L). *Fbxw7*-null IMCD3 cells also displayed elevated levels of *Wnt4* mRNA (Fig. 6M; Table EV1), consistent with the increased SOX9 expression in these cells (Fig. 6C,D). These observations suggest that loss of FBW7 results in activation of the SOX9-WNT4 axis, which plays a key role in developing fibrosis. Consistently, serum BUN, Creatinine, and Cystatin C representing kidney function were also improved in *Cdh16Cre;Fbxw7^{f/f};Sox9^{+/f}* animals (Fig. 6N–P). No significant changes were observed in 2KW/BW (Fig. 6Q), cystic index, cell death (Appendix Fig. S10), or TMEM237 expression (Appendix Fig. S11; Table EV1) in *Cdh16Cre;Fbxw7^{f/f};Sox9^{+/f}* animals compared to *Cdh16Cre;Fbxw7^{f/f}* animals, ruling out effects of SOX9 upregulation on cystogenesis, tubular degeneration or downregulation of TMEM237, at least, as early as its effects on SOX9-driven fibrosis. These findings suggest that the SOX9-WNT4 axis plays a crucial role in fibrosis and kidney function decline in our *Fbxw7*-deletion-based NPHP-like mouse model.

To determine whether our findings extend to human NPHP, we examined SOX9 expression in a patient with an *NPHP1* mutation, the most common genetic cause of juvenile–adult NPHP. Notably, SOX9 levels were increased in kidney tubules across the biopsy compared to controls (Fig. 7A,B). To assess whether the upregulation of SOX9 is also seen in mouse models associated with ciliary defects, a feature often seen in NPHP, we analyzed SOX9

expression in 9-month-old *CaggCre^{ERT2};Ift88^{f/f}* mice, which show stunt cilia, slow-onset cystic disease, and mild fibrosis (Davenport et al, 2007). Consistently, SOX9 levels were significantly increased in the kidney tubules of *CaggCre^{ERT2};Ift88^{f/f}* mice compared to *Ift88^{f/f}* mice (Fig. 7C,D). These findings suggest that enhanced expression of SOX9 is a feature of human NPHP and possibly other forms of NPHP-like pathologies associated with structural defects of primary cilia.

# Discussion

Fibrosis is a hallmark of many cystic kidney diseases, but its underlying molecular mechanisms and impact on kidney function are not well understood. Here, we introduce a new *Fbxw7* knockout mouse model that accurately represents the juvenile–adult form of NPHP. This model displays slow-progressing corticomedullary cysts, tubular degeneration, excessive tubulointerstitial fibrosis, inflammation, and ciliary defects, all associated with reduced kidney function—core features of NPHP. Our findings reveal that the abnormal upregulation of the SOX9-WNT4 axis drives fibrosis, which determines the progressive loss of kidney function seen in NPHP, independent of cystic expansion and tubular deterioration. This study underscores fibrosis as a primary driver of kidney function in NPHP-like pathologies.

NPHP is clinically categorized into three types: infantile, juvenile, and adult, with juvenile and adult NPHP being the most common forms (Srivastava et al, 2017; Wolf and Hildebrandt, 2011; Wolf et al, 2024). However, existing animal models based on *Nphp1* deletion fail to fully replicate the progression of NPHP pathology and early kidney function decline observed in juvenile–adult NPHP patients (Jiang et al, 2008; Kishimoto et al, 2017; Konig et al, 2017; Li et al, 2021; Louie et al, 2010). Our *Fbxw7* knockout mouse model closely mimics the early onset and progression of NPHP seen in juvenile and adult patients, with significant renal pathology and functional decline evident as early as 1-3 months of age. This is a significant improvement over *Nphp1* mouse models (Jiang et al, 2008; Li et al, 2021), where major pathological features, such as cysts, tubular degeneration, or fibrosis, associated with kidney function decline only becoming mildly noticeable after extensive aging. The advantage of our model lies not only in its accurate

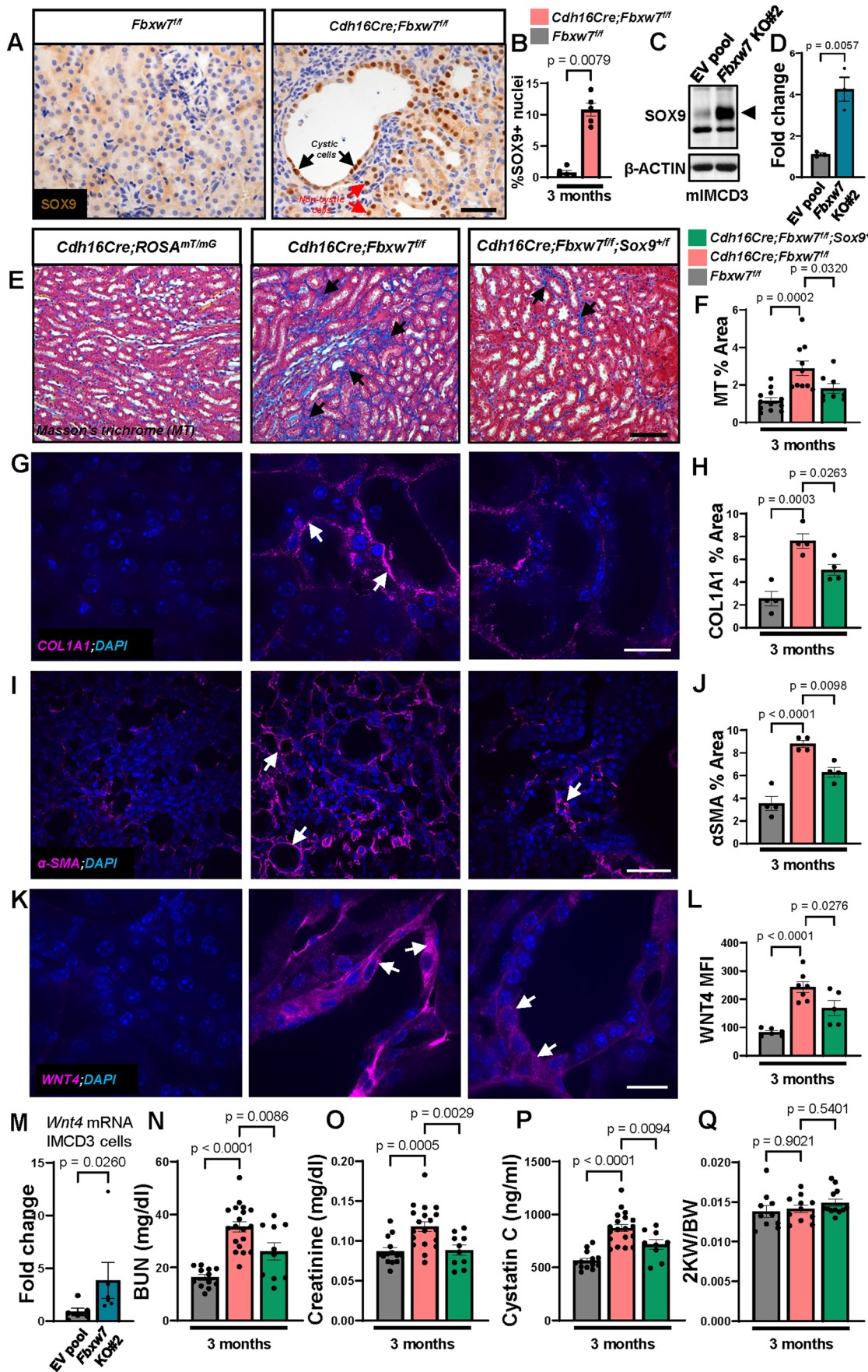

Figure 6.   Increased expression of SOX9 contributes to tubulointerstitial fibrosis and renal function upon loss of FBW7.

(A, B) SOX9 staining and quantification from 3-month-old kidneys of *Fbxw7<sup>f/f</sup>* and *Cdh16Cre;Fbxw7<sup>f/f</sup>* mice. (A) The black arrows show cystic cells, and the red arrows show non-cystic cells with nuclear SOX9 staining (brown). Nuclei are stained with Hematoxylin (blue). Scale bar: 200 μm. (B) Each data point represents the percentage of SOX9+ cells scored per animal (n = 5) from corticomedullary and cortex regions. Statistical analysis was performed using the Mann–Whitney test and is presented as the mean ± SEM. (C, D) Immunoblot and quantification of SOX9 from whole cell lysates of EV pool and *Fbxw7*-null (*Fbxw7 KO#2*) mIMCD3 cells from n = 3 experiments. (E–L) Representative images and quantification of (E, F) MT, (G, H) COL1A1, (I, J) α-SMA, and (K, L) WNT4 staining from 3-month-old kidneys of *Fbxw7<sup>f/f</sup>*, *Cdh16Cre;Fbxw7<sup>f/f</sup>*, and *Cdh16Cre;Fbxw7<sup>f/f</sup>;Sox9<sup>+/f</sup>* mice. Black arrows show (E) MT staining (purple). Scale bar: 200 μm. White arrows show (G) COL1A1, (I) α-SMA, and (K) WNT4 staining (pink). Nuclei are stained with DAPI (blue). Scale bar: (G, K) 20 μm, (I) 50 μm. Each data point represents the percent area of (F) MT (n ≥ 8) staining, (H) COL1A1+ (n = 4) or (J) α-SMA+ (n = 4) percent area, and (L) an average of WNT4 (n ≥ 5) MFI per animal. Statistical analysis was performed using one-way ANOVA followed by Šídák's multiple comparisons test and is presented as the mean ± SEM. (M) qPCR of *Wnt4* mRNA from EV pool and *Fbxw7*-null (*Fbxw7 KO#2*) mIMCD3 cells from n = 6 experiments. (N) Serum BUN (n ≥ 10), (O) Creatinine (n ≥ 10), (P) Cytastin C (n ≥ 9), and (Q) 2KW/BW (n ≥ 10) from 3-month-old *Fbxw7<sup>f/f</sup>*, *Cdh16Cre;Fbxw7<sup>f/f</sup>*, and *Cdh16Cre;Fbxw7<sup>f/f</sup>;Sox9<sup>+/f</sup>* mice. Each data point represents one animal. Statistical analysis was performed using one-way ANOVA followed by Šídák's multiple comparisons test and is presented as the mean ± SEM. Source data are available online for this figure.

replication of NPHP pathology but also in our understanding of FBW7's molecular function (Shimizu et al, 2018). This knowledge allows us to pinpoint the exact molecular player(s) contributing to the various pathologies observed in NPHP and to understand how these pathologies, whether independently or combined, drive kidney function decline. Therefore, our model could serve as a valuable tool for studying juvenile–adult forms of NPHP as well as for evaluating potential therapeutic treatments.

We show that *Fbxw7* deletion led to shorter cilia in mIMCD3 cells, aligning with its known role as a negative regulator of cilia and mirroring the effects seen with loss-of-function mutations in NPHP-causing genes (Srivastava et al, 2017; Stokman et al, 1993). Our unbiased proteomic analysis further supported this observation, which showed a significant reduction in TMEM237 following *Fbxw7* deletion in mIMCD3 cells. In vivo, *Fbxw7* mutant animals exhibited shorter cilia and downregulation of TMEM237 in cystic tubules located in the corticomedullary region, corroborating our findings from *Fbxw7* null mIMCD3 cell lines. Altogether, these results suggest that the shorter cilia and NPHP pathology seen upon loss of FBW7, at least in part, is due to the decreased levels of TMEM237. This aligns with previous findings where loss-of-function mutations in *TMEM237* result in shorter cilia and NPHP pathology in Joubert Syndrome patients (Fleming et al, 2017; Huang et al, 2011). Since we observed a reduction in TMEM237 levels following FBW7 loss, it is unlikely that TMEM237 is a direct target of FBW7, as direct targets would accumulate and show increased protein levels in the absence of FBW7. Uncovering the exact mechanism by which FBW7 regulates TMEM237 expression is a limitation of this study and will be the subject of future investigation.

In our mouse model, fibrosis manifested more extensively than cyst formation or tubular degeneration. By three months of age, mild cysts were evident in the corticomedullary region, surrounded by fibrotic tissue, with additional fibrosis occurring that was not associated with cystic dilations. By seven and ten months of age, these phenotypes were significantly exacerbated, with cysts and tubular degeneration becoming more pronounced alongside widespread fibrosis throughout the kidney. This observation aligns with clinical data from patients and animal models with NPHP-causing gene mutations, where fibrosis is often detected before or in greater abundance than cysts or tubular degeneration (Attanasio et al, 2007; Hoshino et al, 2022; Li et al, 2021; Li et al, 2023). By taking advantage of our knowledge of FBW7's molecular function, we discovered that SOX9, but not c-MYC, served as a crucial regulator of fibrosis in our mouse model. SOX9 regulates fibrosis in extra-

renal tissues and in kidney regeneration following injury (Aggarwal et al, 2024; Kang et al, 2016; Kumar et al, 2015; Raza et al, 2021). Studies have shown that persistent activation of SOX9, unlike its transient activation, triggers a fibrotic response in regenerating kidney tubules (Aggarwal et al, 2024; Kumar et al, 2015). Since FBW7 targets SOX9 for degradation, the loss of FBW7 leads to the accumulation of SOX9, mimicking the effects of its prolonged activation seen in regenerating kidney tubules, albeit to a lesser extent. This could explain why the fibrotic response we observe in FBW7-deficient kidneys is progressive and slower compared to the acute and dramatic fibrosis seen during kidney regeneration after injury. Additionally, SOX9 activation induces the expression of WNT4, which drives fibrosis by activating stromal cells in the surrounding microenvironment (Aggarwal et al, 2024; Kumar et al, 2015). Our findings align with this mechanism, as we observed increased expression of WNT4 and α-SMA, suggesting that a SOX9-WNT4 pathway is involved in the fibrotic response when FBW7 is lost. This was further supported by the observation that reducing SOX9 levels through the deletion of one *Sox9* allele in *Fbxw7* mutant mice ameliorated WNT4 expression and fibrosis, as well as improved kidney function. Notably, upon loss of SOX9, suppression of WNT4 and fibrosis were seen predominantly near the corticomedullary and cortex regions. This is coherent with the fact that we observed different degrees of fibrosis throughout the kidney, but the SOX9 upregulation was mainly apparent in the cortical-medullary and cortex regions. The fact that fibrosis was observed in areas that were not associated with SOX9-positive tubules, especially in the medulla, suggests that other mechanisms, such as the TGF-β pathway/SMAD3, may also contribute to fibrosis, possibly via signals from degenerating tubules and/or F4/80+ macrophages. F4/80+ macrophages are known to contribute to kidney inflammation and fibrosis by promoting pro-inflammatory cytokine release, extracellular matrix deposition, and fibroblast activation (Black et al, 2019; Quatredeniers et al, 2022; Sears et al, 2022; Wen et al, 2021). Nevertheless, the overall reduced fibrosis with significant improvement in kidney function upon loss of SOX9 suggests that SOX9-WNT4-driven fibrosis plays a crucial role in regulating kidney function in NPHP-like pathology resulting from *Fbxw7* deletion. Moreover, our analysis of human NPHP1 patient biopsy displayed a significant increase of SOX9 in kidney tubules compared to controls, further enhancing the translational applicability of our mouse model and the relevance of SOX9-driven fibrosis in the decline of kidney function in human NPHP.

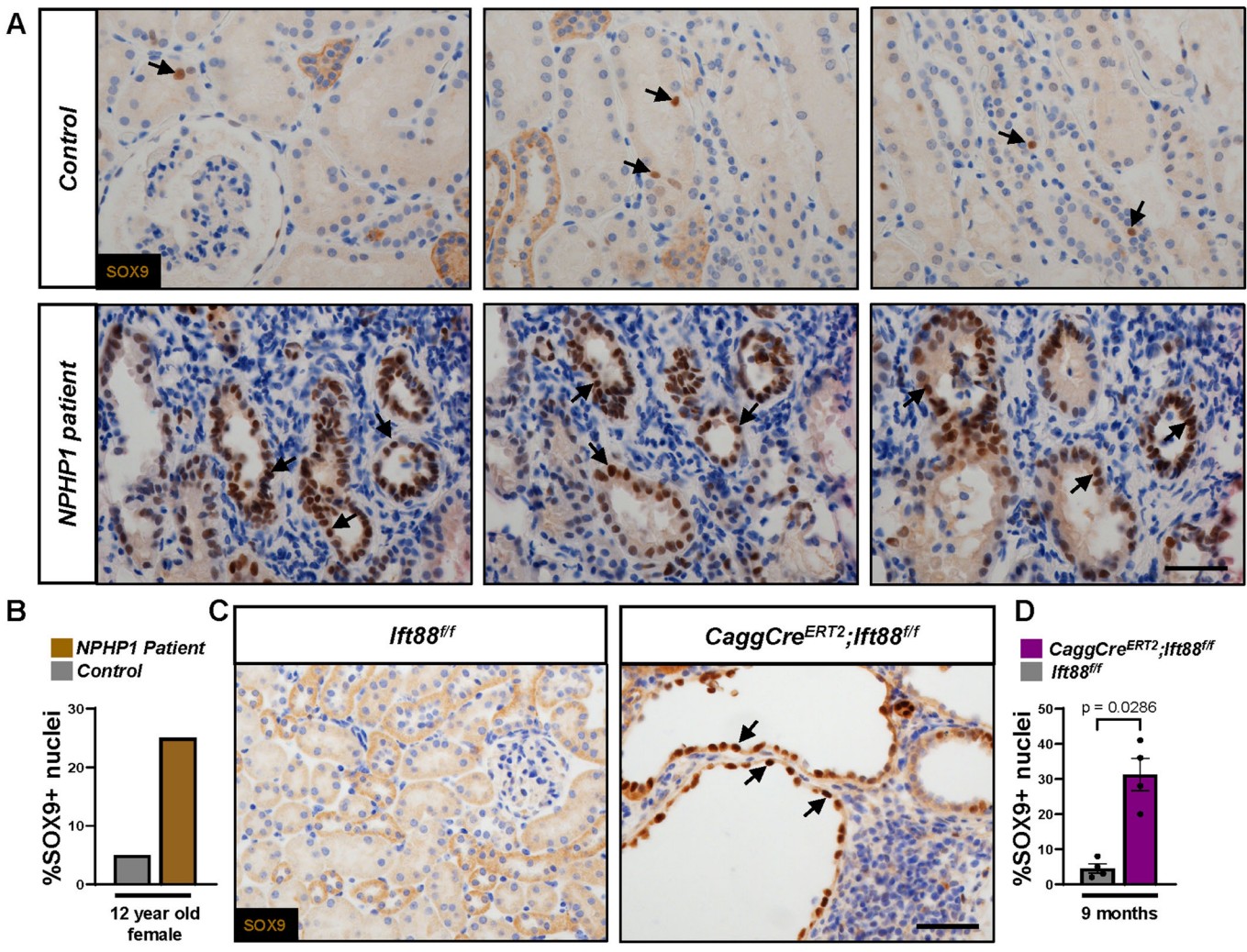

**Figure 7. Increased expression of SOX9 in NPHP patient kidney biopsy and *CaggCre^ERT2^;Ift88^f/f^* mouse model.**

(A, B) SOX9 staining and quantification from a 12-year-old female NPHP patient having a complete deletion of *NPHP1*. Three different images are shown from the same patient biopsy. Black arrows show SOX9+ cells (brown). Nuclei are stained with Hematoxylin (blue). Scale bar: 200 μm. (B) Percentage of SOX9+ cells from control and NPHP1 patient biopsy. (C, D) SOX9 staining and quantification from 9-month-old *Ift88^f/f^* and *CaggCre^ERT2^;Ift88^f/f^* kidneys. Black arrows show SOX9+ cells (brown). Nuclei are stained with Hematoxylin (blue). Scale bar: 200 μm. (D) Each data point represents the percentage of SOX9+ cells scored from cystic tubules per animal ($n = 4$). Statistical analysis was performed using the Mann–Whitney test and is presented as the mean ± SEM. Source data are available online for this figure.

Our mouse model joins mouse models with global or kidney deletions of *Glis2* (Attanasio et al, 2007), *Lkb1* (Viau et al, 2018) or *Aatf* (Jain et al, 2019) in producing NPHP-like phenotypes. While there is no evidence for direct protein-protein or functional interactions between any of these proteins and FBW7, they could be mechanistically connected via common protein network(s). As discussed in the below paragraph, a functional connection between FBW7 and AATF via the DNA Damage response (DDR) is possible. GLIS2 functions primarily as a transcriptional repressor (Attanasio et al, 2007). Therefore, it could be possible that GLIS2-target genes bear an FBW7 phosphodegron. Moreover, several proteins have been identified as negative regulators of LKB1, including Aurora A kinase (Zheng et al, 2018), which is a known FBW7 target (Mao et al, 2004). Therefore, it would be interesting to know whether LKB1 activity is reduced in *Fbxw7*-null cells.

Somatic mutations in *FBXW7* are common in various human cancers, including renal cell carcinoma, yet no such mutations have been identified in patients with NPHP or other autosomal recessive cystic kidney diseases (Cerami et al, 2012; Sailo et al, 2019). Notably, all documented loss-of-function mutations in *FBXW7* occur in its substrate recognition domain, resulting in a complete loss of FBW7 function. Since *Fbxw7*-null animals do not survive embryonic development, it is likely that individuals with homozygous germline mutations in *FBXW7* would also not survive past early development, possibly explaining why no cases of *FBXW7* mutations have been reported in NPHP cohorts. Nevertheless, it is particularly compelling to investigate whether NPHP patients who have homozygous or heterozygous mutations in established NPHP-causing genes, such as *NPHP1* or *NPHP4*, and an additional mutated *FBXW7* allele, may experience exacerbated NPHP pathology. This could potentially qualify FBW7 as a disease

modifier in human NPHP. In regard to the function of FBW7 as a tumor suppressor and especially, in modulating the DDR (Lan and Sun, 2021), it is interesting to note that genes with established functions in the DDR have been identified as NPHP candidate genes in people (Choi et al, 2013; Goggolidou et al, 2014; Slaats et al, 2014). Therefore, it is not entirely surprising that the deletion of *Fbxw7* in the kidney results in an NPHP-like pathology. An important avenue for future research is to determine whether NPHP-like pathology predisposes the kidney to tumor development if a secondary genetic hit occurs. This raises the intriguing possibility of a link between ciliopathy-associated kidney diseases and pre-malignant states.

Overall, utilizing a novel *Fbxw7*-deletion-based mouse model, we show, for the first time, that SOX9-WNT4-dependent fibrosis drives kidney function decline in NPHP-like pathology. These findings establish our *Fbxw7*-deletion-based mouse model as a valuable tool for studying NPHP-like pathology and lay the groundwork for developing therapeutic strategies that target SOX9-WNT4-dependent fibrosis in NPHP and potentially other autosomal recessive cystic kidney diseases.

# Methods

### Reagents and tools table

| Reagent/resource | Reference or source | Identifier or catalog number |
|---|---|---|
| **Experimental models** | | |
| *Cdh16Cre* mice | Jackson Labs | Strain #:012237 |
| *Cagg-Cre^ERT2* mice | Zimmerman/ Jackson Labs | Strain #:004682 |
| *Fbxw7^f/f* mice | Jackson Labs | Strain #:017563 |
| *Sox9^f/f* mice | Jackson Labs | Strain #:013106 |
| *c-Myc^f/f* mice | Cai/MMRC/ Jackson Labs | MMRC Strain #032046-JAX |
| *Ift88^f/f* mice | Zimmerman/ Jackson Labs | Strain #:022409 |
| *Rosa^mT/mG* mice | Zimmerman/ Jackson Labs | Strain #:007676 |
| IMCD3 mouse epithelial cell line | ATCC | CRL-2123 |
| HEK293T human cell line | ATCC | CRL-3216 |
| human *NPHP* kidney biopsy | John Sayer | Not applicable |
| **Recombinant DNA** | | |
| Mouse *Fbxw7* CRISPR plasmid | PMID: 34518642 | Not applicable |
| Mouse TMEM237 plasmid | Sino Biological | MG51523 |
| Mouse SOX9 plasmid | Addgene | Plasmid #36979 |
| **Antibodies** | | |
| Acetylated-α-tubulin (611b) | Sigma-Aldrich | T6793 |

| Reagent/resource | Reference or source | Identifier or catalog number |
|---|---|---|
| Ki67 | Abcam | ab15580 |
| AF647-conjugated F4/80 | Invitrogen | MF48021 |
| NKCC2 | ThermoFisher | PA5142445 |
| Phospho-HISTONE H3 | Cell Signaling | 9701 |
| COL1A1 | Cell Signaling | 72026 |
| α-SMOOTH MUSCLE ACTIN | Cell Signaling | 19245 |
| ATP1A1 | Abcam | ab7671 |
| SOX9 | Sigma-Aldrich | AB5535 |
| c-MYC | Cell Signaling | 9402 |
| TMEM237 | ThermoFisher | PA5-63013 |
| Fluorescein-LTA | Vector Laboratories | FL-1321 |
| Cy5-LTA | USBIO | 518482 |
| Rhodamine-DBA | Vector Laboratories | RL-1032-2 |
| FBW7 | Abnova | H00055294-B01P |
| FBW7 | Bethyl | A301-720A |
| FBW7 | Bethyl | A301-721A |
| β-ACTIN | Santa Cruz | sc-47778 |
| **Oligonucleotides and other sequence-based reagents** | | |
| Mouse *Fbxw7* sgRNA: CACCGATGAAGTCTCGCTGGAACTG | PMID: 34518642 | Not applicable |
| qPCR primers | This study | Table EV1 |
| **Chemicals, enzymes, and other reagents** | | |
| Tamoxifen | Sigma-Aldrich | T5648 |
| Mouse/Rat Cystatin C Quantikine ELISA Kit | Bio-Techne Corporation | MSCTC0 |
| Urea Nitrogen (BUN) Colorimetric Detection Kit | Invitrogen | EIABUN |
| One-step TUNEL In Situ Apoptosis Kit (Red, Elab Fluor® 647) | elabscience | E-CK-A324 |
| ProLong™ Diamond Antifade Mountant with DAPI | Invitrogen™ | P36971 |
| DMEM/F-12, GlutaMAX™ supplement | ThermoFisher Scientific | 10565018 |
| DMEM | Corning | 10-013-CV |
| FBS | Bio-Techne Corporation | S11150 |
| TRYPSIN | ThermoFisher Scientific | 25200056 |
| PBS | Corning | 21-040-CV |
| 4%PFA | ThermoFisher Scientific | J19943.K2 |
| 10%Formalin | Sigma-Aldrich | HT501128 |
| Trizol | ThermoFisher Scientific | 15596026 |
| **Software** | | |
| GraphPad Prism | GraphPad Software, San Diego, CA, USA | V10 |

| Reagent/resource | Reference or source | Identifier or catalog number |
|---|---|---|
| ImageJ | NIH, USA | 1.53k |
| FV10-ASW Viewer | Olympus | 04.02.03.06 |
| OlyVIA | Olympus | 3.4.1 |
| Nikon Elements AR | Nikon | 5.42.04 |
| Nikon Elements Viewer | Nikon | 5.21 |
| Bioquant | Olympus-Bioquant | V23.5.6 |
| Other | | |
| Metabolic cages | Harini Bagavant Lab | Techniplast |

## Mouse models, tissue processing, and serum analysis

Animal housing was done in standard housing conditions with an indoor temperature of around 22 °C and a 12 h light-dark cycle, and experiments were done per the Institutional Animal Care and Use Committee guidelines at the University of Oklahoma Health Science Center (Protocol 22-022 HB). Both sexes were used in all studies. Mouse strains used were *Cdh16Cre, Cagg-Cre^{ERT2}, Fbxw7^{f/f}, Sox9^{f/f}, c-Myc^{f/f}, Ift88^{f/f}, and Rosa^{mT/mG}*. All mice were on C57BL/6-background. For experiments with *Cdh16Cre*, animals were collected at 1-, 3-, 7-, and 10-month time points. For experiments with *Cagg-Cre^{ERT2}*, 8-week-old animals were injected with tamoxifen in a blinded fashion at a dose of 6 mg/40 g of body weight once daily for three consecutive days and collected at 9 months of age. Metabolic cages were used for urine collection over a period of 16 h. The sample size was estimated based on the variability of different assays and stainings used to characterize the NPHP phenotype, including the serum parameters that represent the loss of kidney function. For all experiments, kidneys and body weights were measured. The kidneys were later cut transversally and then fixed in either 10% neutral buffered formalin for paraffin embedding and sectioning or 4% paraformaldehyde for OCT embedding and cryo-sectioning. Tissue processing, embedding, sectioning, and histology stainings were done at the Cell Biology Histology Core and the Stephenson Cancer Center Histology Core of the University of Oklahoma Health Sciences Center (OUHSC). Cystic index and Masson's trichrome staining quantification were done using ImageJ (Schneider et al, 2012; Segard et al, 2024; Song et al, 2022). Blood was collected via cardiac puncture, and serum was collected using serum separator tubes (BD Microtainer 365967). Serum BUN and Cystatin C analysis were done using Invitrogen BUN Colorimetric Detection Kit (Cat. no. EIABUN) and R&D Systems Cystatin C Quantikine ELISA Kit (Cat. no. MSCTC0), respectively. Serum electrolytes were analyzed from frozen serum using Abaxis VetScan VS2. Serum and urine creatinine analysis was done using Liquid chromatography–mass spectrometry (LC-MS) by the Bioanalytical Core, O'Brien Center for Acute Kidney Injury Research at the University of Alabama at Birmingham. Total protein concentration in urine was measured using Pierce™ BCA Protein Assay Kit (Cat. No. 23225).

## Cell culture and generation of Fbxw7-null mIMCD3 cells

HEK293T cells were maintained in Dulbecco's modified Eagle's medium (DMEM) supplemented with 10% fetal bovine serum. mIMCD3 cells were cultured in a 1:1 mixture of DMEM and Ham's F-12,

supplemented with 10% FBS. HEK293T and mIMCD3 authenticated cells were bought commercially and tested for mycoplasma routinely. mIMCD3 cells underwent transfection using a lentivirus carrying a vector that encodes Cas9. This vector was either an empty vector (EV) or included a single guide RNA (sgRNA) specifically targeting *Fbxw7* (sequence: 5'-CACCGATGAAGTCTCGCTGGAACTG-3'). The chosen sgRNA sequence and corresponding vector were previously validated for their efficiency in knocking out *Fbxw7* in mouse cell lines (Petsouki et al, 2021). Following 24 h post-transduction, the cells were subjected to puromycin treatment at a concentration of 2 μg/ml for 48 h. Subsequently, to initiate single-cell-based colonies, the cells were dissociated with trypsin and individually sorted into 96-well plates using FACS. Several single-cell-based clones were then expanded and assessed for the absence of FBW7 through Western blot analysis. We successfully identified two clones with complete elimination of FBW7 protein, *Fbxw7*-KO #2 and *Fbxw7*-KO#7. Both parental mIMCD3 cells and those stably transfected with the empty vector were used as controls for comparative purposes.

## Immunofluorescence, TUNEL, and microscopy

For immunofluorescence studies, 10 μm or 50 μm kidney cryosections or mIMCD3 cells serum starved for 48 h on glass coverslips were used. Sections were washed three times with PBS, 5 min each, and then incubated in blocking solution (3% BSA, 0.15% Triton X-100, 5% Goat Serum in PBS) for 1 h at room temperature. Samples were then incubated in primary antibody overnight at 4 °C, washed with PBS three times 5 min each, and then incubated with the appropriate secondary antibodies in a blocking solution for 1 h at room temperature. Sections were then washed with PBS 3 times for 5 min each, mounted with Diamond DAPI, and visualized with confocal microscopy. Antibodies used were acetylated-α-tubulin (611b, 1:2000; Sigma-Aldrich), Ki67 (1:200; ab15580), AF647-conjugated F4/80 (1:50; Invitrogen MF48021), NKCC2 (1:200, ThermoFisher PA5142445), Phospho-HISTONE H3 (1:200, Cell Signaling 9701), COL1A1 (1:200, Cell Signaling 72026), α-SMOOTH MUSCLE ACTIN (1:200, Cell Signaling 19245), ATP1A1 (1:500, abcam ab7671), SOX9 (1:200, Sigma-Aldrich AB5535), c-MYC (1:200, Cell Signaling 9402), and TMEM237 (1:200, ThermoFisher PA5-63013). Fluorescein (1:1000; Vector Laboratories FL-1321) or Cy5 (1:1000; USBIO 518482) conjugated Lotus Tetragonolobus Lectin (LTA) and Rhodamine conjugated Dolichos Biflorus Agglutinin (DBA) (1:1000; Vector Laboratories RL-1032-2), was added to co-stain proximal tubules and collecting ducts, respectively, in the secondary antibody solution wherever needed. For EdU incorporation, mice received 50 mg per kg of body weight of EdU (Invitrogen, catalog no. A10044) by intraperitoneal injection 4–6 h before euthanasia. EdU staining was done using Click-iT EdU Alexa Fluor-594 Imaging Kit (Invitrogen, catalog no. C10339). For TUNEL staining, the One-step TUNEL Apoptosis Kit was used per the manufacturer's instructions (Elabscience E-CK-A324). All images were obtained either by an Olympus FV1000 confocal microscope, Nikon CSU-W1 SoRa Spinning Disk confocal microscope, Olympus BX51, or Olympus VS120 Virtual Slide System.

## Immunoblotting and quantitative polymerase chain reaction (qPCR)

For FBW7 blots from mIMCD3 cells, Abnova FBW7 antibody (1:100, Abnova H00055294-B01P) was used to pull down FBW7

and Bethyl FBW7 antibodies (1:1000 in a 1:1 ratio, Bethyl A301-720A, and A301-721A) were used to detect FBW7 levels in controls and *Fbxw7*-null single-cell clones. Other antibodies used were SOX9 (1:1000, Sigma-Aldrich AB5535), TMEM237 (1:1000, ThermoFisher PA5142445), c-MYC (1:1000, Cell Signaling 9402), and β-ACTIN (1:1000, sc-47778). Densitometric quantification was performed with ImageJ. RNA was isolated from cells using Trizol (Invitrogen) and was further cleaned up using RNeasy mini kit (Qiagen). Purified RNA was transcribed into cDNA using Maxima First Strand cDNA Synthesis Kit (K1641). Biorad CFX96 Touch Real-Time PCR machnie was used for qPCR.

## Human samples

The patient was a 12-year-old female who was admitted to the hospital with abdominal pain and a raised serum creatinine. Following informed consent, a kidney biopsy was performed, and the findings were consistent with nephronophthisis. Genetic investigations confirmed a homozygous *NPHP1* whole gene deletion. The collection of clinical data was undertaken with informed consent, and ethical approval was given by the National Research Ethics Service Committee North East—Newcastle and North Tyneside 1 (08/H0906/28 + 5) and the National Research Ethics Service (NRES) Committee North East (14/NE/1076). Experiments conformed to the principles set out in the WMA Declaration of Helsinki and the Department of Health and Human Services Belmont Report. Genetic investigations were performed by the Northern Genetics Service diagnostics laboratory using Multiplex ligation-dependent probe amplification (MLPA).

## Proteomic screen in mIMCD3 cells

The total protein from each sample was reduced, alkylated, and purified by chloroform/methanol extraction prior to digestion with sequencing grade-modified porcine trypsin (Promega). Tryptic peptides were then separated by reverse phase XSelect CSH C18 2.5 um resin (Waters) on an in-line $150 \times 0.075$ mm column using an UltiMate 3000 RSLCnano system (Thermo). Peptides were eluted using a 60 min gradient from 98:2 to 65:35 buffer A:B ratio. Eluted peptides were ionized by electrospray (2.2 kV) followed by mass spectrometric analysis on an Orbitrap Exploris 480 mass spectrometer (Thermo). To assemble a chromatogram library, six gas-phase fractions were acquired on the Orbitrap Exploris with 4 $m/z$ DIA spectra (4 $m/z$ precursor isolation windows at 30,000 resolution, normalized AGC target 100%, maximum inject time 66 ms) using a staggered window pattern from narrow mass ranges using optimized window placements. Precursor spectra were acquired after each DIA duty cycle, spanning the $m/z$ range of the gas-phase fraction (i.e., 496–602 $m/z$, 60,000 resolution, normalized AGC target 100%, maximum injection time 50 ms). For wide-window acquisitions, the Orbitrap Exploris was configured to acquire a precursor scan (385–1015 $m/z$, 60,000 resolution, normalized AGC target 100%, maximum injection time 50 ms) followed by $50 \times 12$ $m/z$ DIA spectra (12 $m/z$ precursor isolation windows at 15,000 resolution, normalized AGC target 100%, maximum injection time 33 ms) using a staggered window pattern with optimized window placements. Precursor spectra were acquired after each DIA duty cycle. To assemble a data-dependent spectral library, a pooled

### The paper explained

#### Problem

Fibrosis is a hallmark of many cystic kidney diseases, particularly autosomal recessive disorders like nephronophthisis (NPHP). Despite its strong correlation with kidney function decline, the causal role of fibrosis and its underlying molecular mechanisms remain poorly understood.

#### Results

We developed an *Fbxw7*-deletion-based mouse model that faithfully mimics a juvenile–adult NPHP-like pathology characterized by cortico-medullary cysts, tubular degeneration, and excessive fibrosis, along with progressive kidney function decline. Utilizing this model, we identified SOX9 as a central regulator of fibrosis, revealing fibrosis—not cystogenesis or tubular degeneration—as the primary driver of kidney function loss in NPHP-like disorders.

#### Impact

This study identifies fibrosis as the principal driver of kidney function decline in NPHP. Our findings provide critical insights into the molecular mechanisms of NPHP and highlight SOX9-dependent fibrosis as a potential therapeutic target for treating NPHP and potentially other autosomal recessive kidney disorders.

peptide reference samples was separated into 46 fractions on a $100 \times 1.0$ mm Acquity BEH C18 column (Waters) using an UltiMate 3000 UHPLC system (Thermo) with a 50 min gradient from 99:1 to 60:40 buffer A:B ratio under basic pH conditions, then consolidated into 18 super-fractions. Each super-fraction was further separated by reverse phase XSelect CSH C18 2.5 um resin (Waters) on an in-line $150 \times 0.075$ mm column using an UltiMate 3000 RSLCnano system (Thermo). Peptides were eluted using a 60 min gradient from 98:2 to 65:35 buffer A:B ratio. Eluted peptides were ionized by electrospray (2.2 kV) followed by mass spectrometric analysis on an Orbitrap Exploris 480 mass spectrometer (Thermo). MS data were acquired using the FTMS analyzer in profile mode at a resolution of 120,000 over a range of 375–1500 $m/z$. Following HCD activation, MS/MS data were acquired using the FTMS analyzer in centroid mode at a resolution of 15,000 and normal mass range with normalized collision energy of 30%. Buffer A = 0.1% formic acid, 0.5% acetonitrile. Buffer B = 0.1% formic acid, 99.9% acetonitrile. Both buffers were adjusted to pH 10 with ammonium hydroxide for offline separation.

Following data acquisition, data were searched using an empirically corrected library against the UniProt Mus musculus database (April 2022) and a quantitative analysis was performed to obtain a comprehensive proteomic profile. Proteins were identified and quantified using EncyclopeDIA (Searle et al, 2018) and visualized with Scaffold DIA using 1% false discovery thresholds at both the protein and peptide level. Protein-exclusive intensity values were assessed for quality using Protei-Norm (Graw et al, 2020). The data was normalized using cyclic loess and statistical analysis was performed using linear models for microarray data (limma) with empirical Bayes (eBayes) smoothing to the standard errors (Ritchie et al, 2015). Proteins with an FDR-adjusted $P$ value <0.05 and a fold change >2 were considered significant.

## Statistics

GraphPad Prism was used for all statistical analyses. Quantitative results that required comparisons between groups were subjected to statistical analysis using a *t* test for two groups, one-way ANOVA for multiple groups, or two-way ANOVA for multiple groups with different time points, followed by an appropriate ad hoc test. Serum analysis and quantifications were done blindly wherever possible.

## Graphics

The synopsis was created using BioRender.com.

## Data availability

The proteomics data have been deposited to the ProteomeXchange Consortium via the PRIDE partner repository (https://www.ebi.ac.uk/pride/archive/projects/PXD061542). Source data for images have been submitted to BioImage Archive (https://www.ebi.ac.uk/biostudies/bioimages/studies/S-BIAD1704?key=404186c2-0aee-4454-80d9-6f0451acf532).

The source data of this paper are collected in the following database record: biostudies:S-SCDT-10_1038-S44321-025-00233-3.

## Peer review information

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

## Acknowledgements

The authors thank the Cell Biology Histology Core Facility and the Histology and Immunohistochemistry Core of the University of Oklahoma Health Sciences Center (OUHSC) Stephenson Cancer Center for histology and tissue sectioning, supported by P30CA225520 and P20GM103639. The authors thank the IDeA National Resource for Quantitative Proteomics and NIH/NIGMS grant R24GM137786 for quantitative proteomics and analysis. The authors thank Drs. Christine Vanbeek for advice on histopathological analyses; Harini Bagavant for providing metabolic cages; and Ben Cowley for advice on metabolic acidosis. This work was supported by R01DK126705 and R01DK138339 from the NIH/NIDDK and the John S Gammill Endowed Chair in Polycystic Kidney Disease to LT. JAS is supported by Kidney Research UK (Paed_RP_001_20180925); LifeArc (LifeArc Pathfinder 1.0.), Medical Research Council (MR/Y007808/1). KAZ is supported by NIH/NIDDK grants K01DK119375 and R01DK129255.

## Author contributions

**Maulin Mukeshchandra Patel**: Conceptualization; Data curation; Formal analysis; Investigation; Methodology; Writing—original draft; Writing—review and editing. **Vasileios Gerakopoulos**: Data curation; Formal analysis. **Bryan Lettenmaier**: Data curation. **Eleni Petsouki**: Data curation. **Kurt A Zimmerman**: Resources. **John A Sayer**: Resources. **Leonidas Tsiokas**: Conceptualization; Resources; Formal analysis; Supervision; Funding acquisition; Investigation; Project administration; Writing—review and editing.

Source data underlying figure panels in this paper may have individual authorship assigned. Where available, figure panel/source data authorship is listed in the following database record: biostudies:S-SCDT-10_1038-S44321-025-00233-3.

## Disclosure and competing interests statement

The authors declare no competing interests.

# Expanded View Figures

**Figure EV1. Deletion of *Fbxw7* results in cyst formation in different tubular segments of the kidney.**

(**A**) Cystic index represented as the percentage of cystic area per kidney section from 1-, 3-, 7-, and 10-month-old *Fbxw7^{f/f}* and *Cdh16Cre;Fbxw7^{f/f}* mice. Each data point represents one animal (n ≥ 3). Statistical analysis was performed using two-way ANOVA and is presented as the mean ± SEM. (**B**) Representative images of LTA+ and mG + cystic tubules in *Cdh16Cre;ROSA^{mT/mG};Fbxw7^{f/f}* (middle and right panels) and *Cdh16Cre;ROSA^{mT/mG}* (left panel) mice. White arrows show cystic mG+ tubules that are LTA +. Nuclei are stained with DAPI (blue). Scale bar: 50 μm. (**C–J**) Representative images of immunofluorescence staining for (**C, D**) proximal tubule marker, Lotus tetragonolobus agglutinin (LTA), (**E, F**) thick ascending limb of the loop of Henle marker, NKCC2, and (**G, H**) collecting duct marker, Dolichos biflorus agglutinin (DBA) from 3- and 7-month-old kidneys of *Fbxw7^{f/f}* and *Cdh16Cre;Fbxw7^{f/f}* mice. White arrows show dilated or cystic tubular segments of the kidney that are positive for (**C, D**) LTA (green), (**E, F**) NKCC2 (pink), and (**G, H**) DBA (red). Nuclei are stained with DAPI (blue). Scale bar: 50 μm. (**I, J**) Pie chart displaying the percentage of cystic tubules distributed across different kidney segments from 3- and 7-month-old *Fbxw7^{f/f}* and *Cdh16Cre;Fbxw7^{f/f}* mice (n ≥ 3).

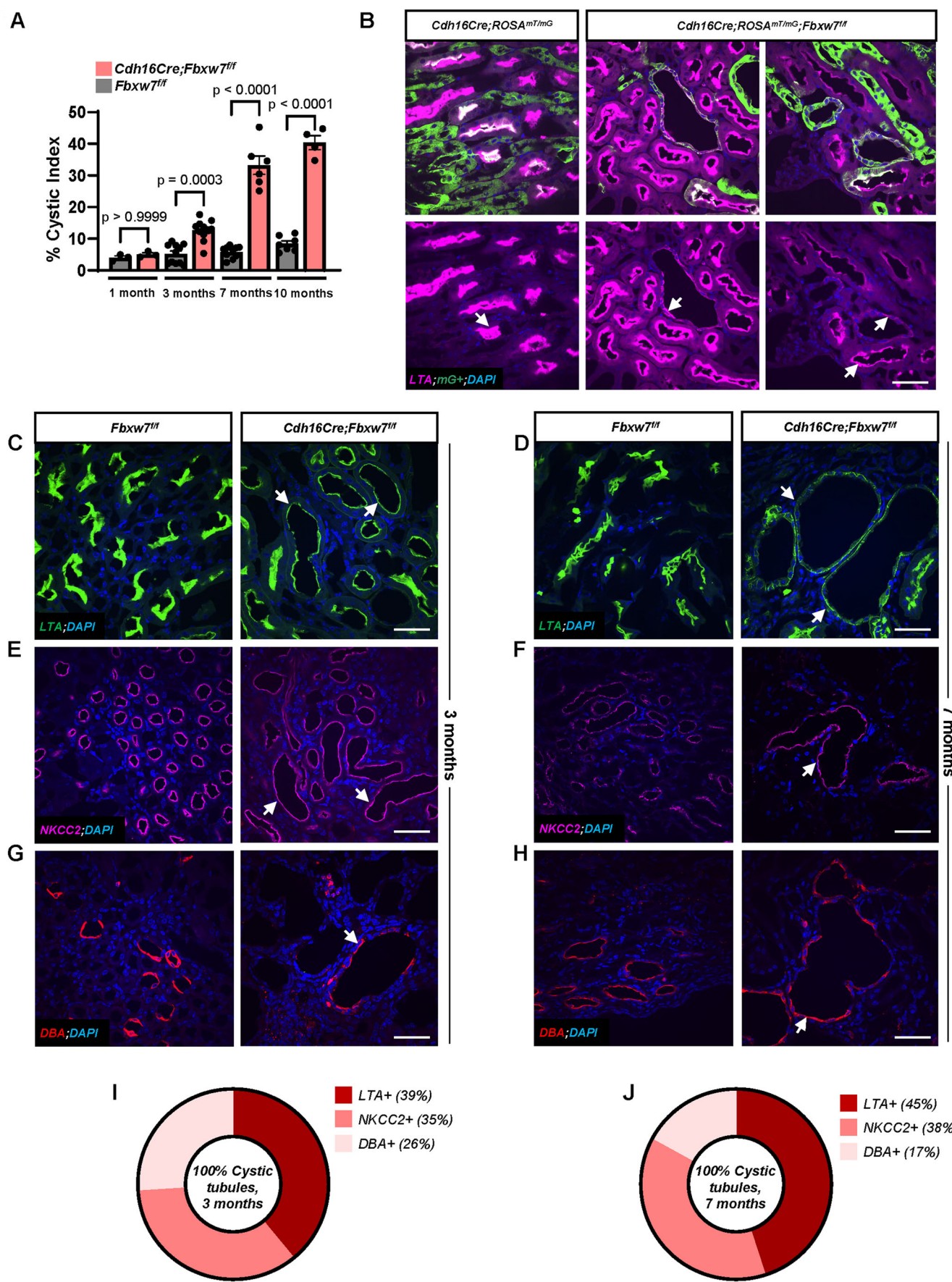

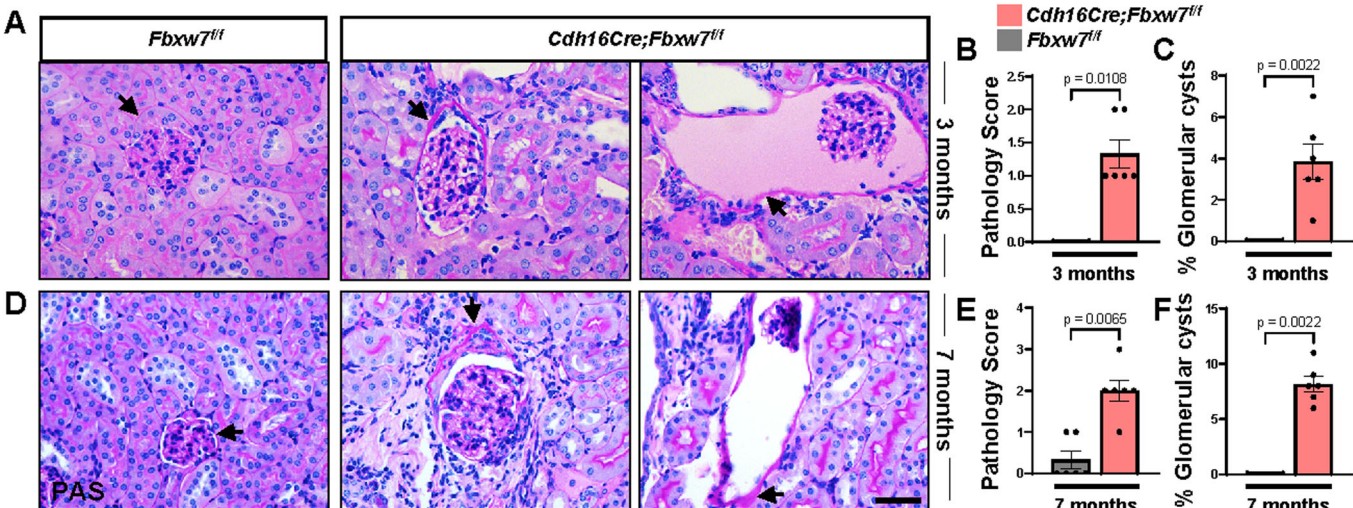

**Figure EV2. Deletion of *Fbxw7* results in modest glomerular cysts and Bowman's capsule thickening.**

(A–F) Representative images and quantification of glomerular cysts and mild thickening of Bowman's capsule (black arrows) using PAS staining from 3- and 7-month-old kidneys of *Fbxw7^{f/f}* and *Cdh16Cre;Fbxw7^{f/f}* mice. Nuclei are stained with Hematoxylin (blue). Scale bar: 200 μm. (B, E) Pathology score for the mild thickening of Bowman's capsule and (C, F) percentage of glomerular cysts scored per animal (n = 6). Statistical analysis was performed using the Mann–Whitney test and is presented as the mean ± SEM.

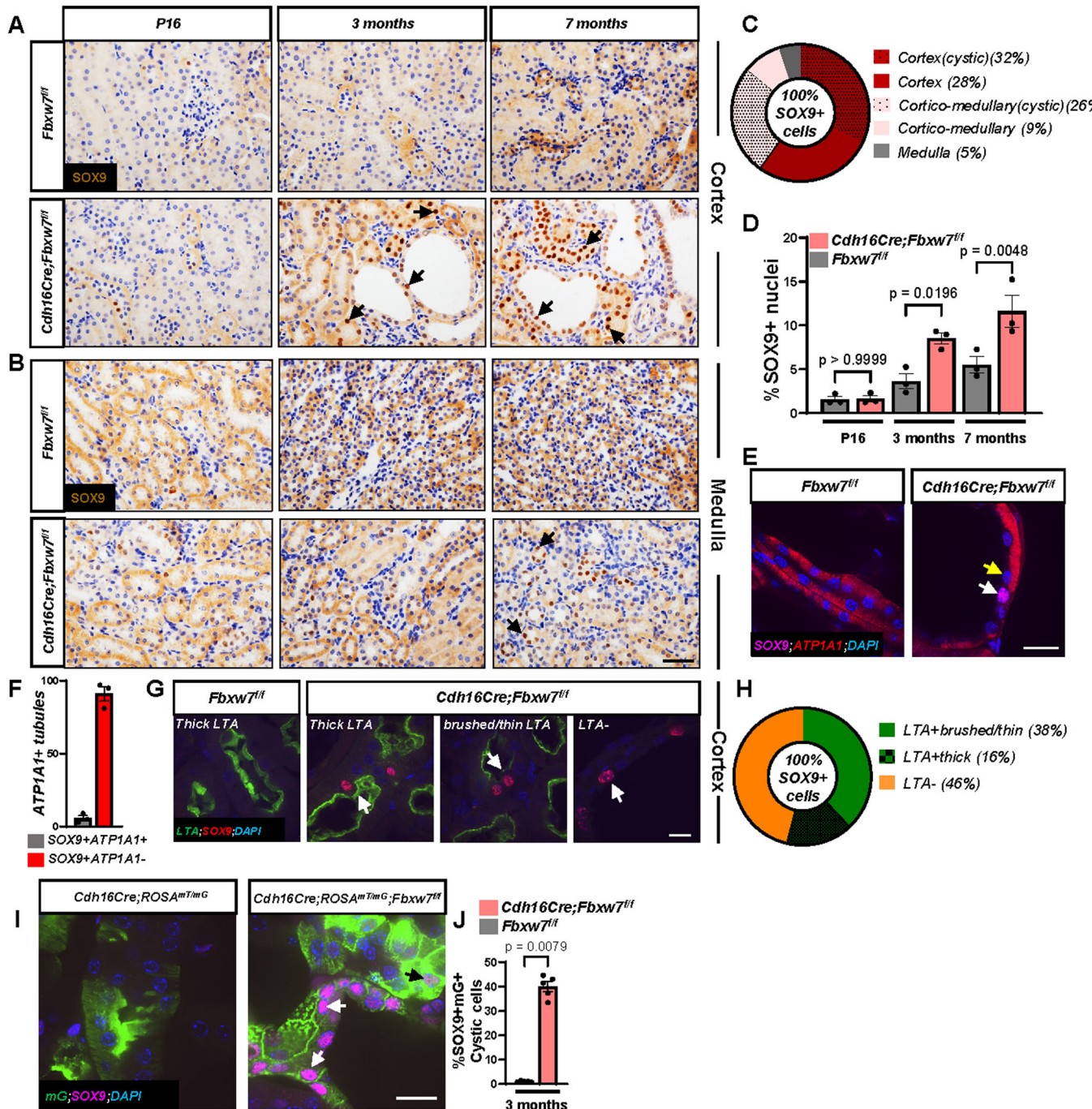

**Figure EV3.  Spatio-temporal characterization of increased SOX9 expression in *Fbxw7*-deletion-based NPHP-like mouse model.**

(A–D) SOX9 staining and quantification from different kidney regions from P16-, 3-, and 7-month-old *Fbxw7^{f/f}* and *Cdh16Cre;Fbxw7^{f/f}* mice. (A, B) Black arrows show SOX9+ cells (brown). Nuclei are stained with Hematoxylin (blue). Scale bar: 100 μm. (C) Pie chart displaying the percentage SOX9+ cells distributed across different kidney regions from 3-month-old *Fbxw7^{f/f}* and *Cdh16Cre;Fbxw7^{f/f}* mice. (D) Each data point represents the percentage of SOX9+ cells scored per animal from P16-, 3-, and 7-month-old *Fbxw7^{f/f}* and *Cdh16Cre;Fbxw7^{f/f}* mice (*n* = 3). Statistical analysis was performed using two-way ANOVA and is presented as the mean ± SEM. (E, F) Representative images and quantification of SOX9+ cells within ATP1A1+ tubules that have lost ATP1A1 expression. (E) The white arrow shows the SOX + ATP1A1- cell, and the yellow arrow shows the SOX9-ATP1A1+ cell within the same cystic tubule. Nuclei are stained with DAPI (blue). Scale bar: 20 μm. (F) Each data point represents the percentage of SOX9+ cells scored per animal from ATP1A1+ tubules of 3-month-old *Cdh16Cre;Fbxw7^{f/f}* mice. (G, H) Representative images and distribution of SOX9+ cells in LTA+normal-looking, LTA+brushed proximal tubules, and LTA- tubules. (G) The white arrow shows SOX9+ cells in respective tubules. Nuclei are stained with DAPI (blue). Scale bar: 20 μm. (I, J) SOX9 staining and quantification from 3-month-old kidneys of *Cdh16Cre;ROSA^{mT/mG}* and *Cdh16Cre;ROSA^{mT/mG};Fbxw7^{f/f}* mice. (I) The white and black arrows show nuclear SOX9 staining (pink) in mG+ cystic and non-cystic tubules (green), respectively. Nuclei are stained with DAPI (blue). Scale bar: 20 μm. (J) Each data point represents the percentage of SOX9+ cells scored from mG+ cytic tubules per animal (*n* = 5). Statistical analysis was performed using the Mann–Whitney test and is presented as the mean ± SEM.

