## [Peer Review File · EMBO Molecular Medicine]

SOX9-dependent fibrosis drives renal function in nephronophthisis

Maulin Patel, Vasileios Gerakopoulos, Bryan Lettenmaier, Eleni Petsouki, Kurt Zimmerman, John Sayer, and Leonidas Tsiokas

Corresponding author: Leonidas Tsiokas (ltsiokas@ouhsc.edu)

Review Timeline:

Submission Date:	18th Oct 24
Editorial Decision:	12th Nov 24
Revision Received:	16th Feb 25
Editorial Decision:	11th Mar 25
Revision Received:	18th Mar 25
Accepted:	26th Mar 25

Editor: Lise Roth

Transaction Report:

12th Nov 2024

Dear Prof. Tsiokas,

Thank you for the submission of your manuscript to EMBO Molecular Medicine. We have now heard back from the referees who reviewed your manuscript. As you will see from the reports below, the referees acknowledge the interest of the study and are overall supporting publication of your work pending appropriate revisions.

All referees' concerns should be addressed, and in particular, a clear link with human NPHP should be provided. On the other hand, applications of the findings to other kidney diseases could be addressed by discussion only.

Acceptance of the manuscript will entail a second round of review. EMBO Molecular Medicine encourages a single round of revision only and therefore, acceptance or rejection of the manuscript will depend on the completeness of your responses included in the next, final version of the manuscript. For this reason, and to save you from any frustrations in the end, I would strongly advise against returning an incomplete revision.

We are expecting your revised manuscript within three months, if you anticipate any delay, please contact us.

We require:

4) A .docx formatted letter INCLUDING the reviewers' reports and your detailed point-by-point responses to their comments. As part of the EMBO Press transparent editorial process, the point-by-point response is part of the Review Process File (RPF), which will be published alongside your paper.

5) A complete author checklist, which you can download from our author guidelines (<https://www.embopress.org/page/journal/17574684/authorguide#submissionofrevisions>). Please insert information in the checklist that is also reflected in the manuscript. The completed author checklist will also be part of the RPF.

6) All Materials and Methods need to be described in the main text using our 'Structured Methods' format. According to this format, the Methods section includes a Reagents and Tools Table (listing key reagents, experimental models, software and relevant equipment and including their sources and relevant identifiers) followed by a Methods and Protocols section describing the methods, ideally using a step-by-step protocol format. The aim is to facilitate adoption of the methodologies across labs. Please download and fill our Reagents and Tools Table template (.docx), which you can find in our author guidelines: <https://www.embopress.org/page/journal/14693178/authorguide#structuredmethods>.

<https://www.embopress.org/doi/10.15252/msb.20178071>

7) It is mandatory to include a 'Data Availability' section after the Materials and Methods. Before submitting your revision, primary datasets produced in this study need to be deposited in an appropriate public database, and the accession numbers and database listed under 'Data Availability'. Please remember to provide a reviewer password if the datasets are not yet public (see <https://www.embopress.org/page/journal/17574684/authorguide#dataavailability>).

In case you have no data that requires deposition in a public database, please state so in this section.

Note that the Data Availability Section is restricted to new primary data that are part of this study.

8) For data quantification: please specify the name of the statistical test used to generate error bars and P values, the number (n) of independent experiments (specify technical or biological replicates) underlying each data point and the test used to calculate p-values in each figure legend. The figure legends should contain a basic description of n, P and the test applied. Graphs must include a description of the bars and the error bars (s.d., s.e.m.). Please provide exact p values.

12) Author contributions: CRedit has replaced the traditional author contributions section because it offers a systematic machine readable author contributions format that allows for more effective research assessment. Please remove the Authors Contributions from the manuscript and use the free text boxes beneath each contributing author's name in our system to add specific details on the author's contribution. More information is available in our guide to authors.

13) Disclosure statement and competing interests: We updated our journal's competing interests policy in January 2022 and request authors to consider both actual and perceived competing interests. Please review the policy <https://www.embopress.org/competing-interests> and update your competing interests if necessary.

14) Every published paper now includes a 'Synopsis' to further enhance discoverability. Synopses are displayed on the journal webpage and are freely accessible to all readers. They include a short stand first (maximum of 300 characters, including space) as well as 2-5 one-sentences bullet points that summarizes the paper. Please write the bullet points to summarize the key NEW findings. They should be designed to be complementary to the abstract - i.e. not repeat the same text. We encourage inclusion of key acronyms and quantitative information (maximum of 30 words / bullet point). Please use the passive voice. Please attach these in a separate file or send them by email, we will incorporate them accordingly.

Please also suggest a visual abstract to illustrate your article as a PNG file 550 px wide x 300-600 px high. A cropped portion of this image will serve as thumbnail for the table of content on our webpage.

15) As part of the EMBO Publications transparent editorial process initiative (see our Editorial at <http://embomolmed.embopress.org/content/2/9/329>), EMBO Molecular Medicine will publish online a Review Process File (RPF) to accompany accepted manuscripts.

In the event of accepted acceptance, this file will be published in conjunction with your paper and will include the anonymous referee reports, your point-by-point response and all pertinent correspondence relating to the manuscript. Let us know whether you agree with the publication of the RPF and as here, if you want to remove or not any figures from it prior to publication. Please note that the Authors checklist will be published at the end of the RPF.

EMBO Molecular Medicine has a "scooping protection" policy, whereby similar findings that are published by others during

review or revision are not a criterion for rejection. Should you decide to submit a revised version, I do ask that you get in touch after three months if you have not completed it, to update us on the status.

I look forward to receiving your revised manuscript.

Yours sincerely,

Lise Roth

***** Reviewer's comments *****

Referee #1 (Comments on Novelty/Model System for Author):

Technical Quality (inc. statistical analysis):

The paper exhibits robust experimental design, well-defined procedures, and strong statistical analysis. Glomerular filtration rate (GFR) measurements or other histological glomerular indicators may be used to analyze the loss of kidney function caused by FBW7 deletion in greater detail.

Novelty:

The present study unveiled the FBW7-SOX9-WNT4 axis as a novel driver of renal fibrosis and critical for most signaling pathways in NPHP. While the involvement of the SOX9-WNT4 axis in fibrosis has been documented in multiple contexts, including during kidney disease, the study underlines its specific contribution to conditions similar to NPHP. The findings provides new opportunities for therapeutic treatments that target fibrosis in NPHP and maybe other kidney illnesses by clarifying the pathways via which FBW7 depletion triggers the activation of this axis.

Medical impact:

The study presents potential insights relevant to autosomal recessive kidney diseases, particularly for fibrosis as a therapeutic target. However, the impact is somewhat limited by the focus on a single model and lack of human tissue validation or genetic studies in patients with NPHP. Future studies, possibly involving patient-derived cell models or organoids, could strengthen the translational potential and increase medical relevance.

Adequacy of Model System: Adequate

The Fbxw7-deficient mouse model successfully simulates a range of NPHP-like symptoms and allows investigation into fibrosis pathways in cystic kidney disease.

Referee #1 (Remarks for Author):

Presented herein is an in-depth analysis of the deletion of Fbxw7 in kidney epithelial cells. It was noted that mice with deleted Fbxw7 express a juvenile-adult NPHP-like phenotype, including severe fibrosis, tubular degeneration, slow-progressing corticomedullary cysts, and progressive loss of renal function. This is in concert with the NPHP characteristic trio and gives important insight into the growing importance of fibrosis in autosomal recessive kidney diseases, thus successfully modeling renal disease. The deletion of Fbxw7 leads to shortened cilia and reduced levels of TMEM237, a known ciliopathy protein. The authors have identified the increased expression of proteins like SOX9 and c-MYC among the post-deletion consequences of Fbxw7. Normalization of c-MYC barely had any effects on pathology, though modulation of SOX9 was able to reduce fibrosis; thus, only a subset of FBW7 substrates seem critical in influencing disease pathology. Overexpression of SOX9 and its downstream effector WNT4 in the Fbxw7-deleted model points to this axis as a critical factor in the development of fibrosis. This is further proven beyond reasonable doubt that heterozygous deletion of Sox9 in compound mutant mice restores the nuclear levels of WNT4 and diminishes fibrosis.

Following are my comments on the study.

- The Fbxw7-deficient NPHP-like model may be further used for studying renal function decline in other kidney diseases; in other words, findings could similarly apply to other kidney cystic diseases, such as ADPKD.

- Although the SOX9-WNT4 axis may be in focus, other classic pro-fibrotic pathways associated with renal fibrosis, including that of TGF- β , are not pursued. This would, at best, diminish the comprehensive value of mechanistic insight and obscure potential pathway interplay that may fully explain NPHP fibrosis.

- The research will pursue renal function impairment by defects in urine concentration as well as through serum markers like BUN, creatinine, and cystatin C. Further work using GFR measurements or histological markers of glomerular injury would support a more advanced understanding of renal function loss in the context of FBXW7 deletion.

-Although the paper points out the increase in F4/80-positive macrophages as an indication of inflammation in kidneys with Fbxw7 deletion, the details are not stated.

Even in NPHP, it is regarded that inflammation is one of the most important factors in the development of renal fibrosis. Inflammatory responses can greatly contribute to the development of renal fibrosis, and further investigation into the link between inflammation and fibrosis may greatly enhance our understanding of the similar pathophysiology as seen in NPHP. -In this study, findings are based on the mouse model alone, and how well these results will translate into human NPHP or other autosomal recessive cystic kidney disorders is not known. Human tissue validation or genetic studies in NPHP patients could strengthen translational applicability.

Referee #2 (Remarks for Author):

This study addresses two major issues in nephronophthisis (NPHP) cystic kidney disease: (1) generating a mouse model that replicates key features of human NPHP, including progressive renal dysfunction, and (2) identifying mechanisms driving fibrosis in genetic cystic kidney disease, which are poorly understood.

The study successfully generated an NPHP-like disease model by deleting FBW7 in Cdh16-expressing nephron epithelia. This led to a SOX9 response, an FBW7-regulated gene, in FBW7-deficient Cdh16+ epithelia, resulting in fibrosis via Wnt4, a short-range secretory ligand. Importantly, partial SOX9 deletion in Cdh16-specific, FBW7-deficient animals reduced fibrosis and promoted renal function recovery, as indicated by significantly lower serum creatinine and cystatin C levels. With respect to NPHP-like disease, an unbiased identification of TMEM237-a gene associated with Joubert syndrome-related disorders (JSRD), which includes cystic kidney disease as one of the manifestations, provide additional supports to the development of a cystic kidney disease-like phenotype in this FBW7-deficient mouse model. One of the hallmarks of kidney disease in JSRD patients are corticomedullary cystic kidneys with exaggerated fibrosis.

Taken together, this study represents a significant advancement in the field. Further, this study is suitable for broad readership because ciliopathies involve different organs, and these findings might have broad implications, for example, in pathological remodeling of tumor microenvironment in FBXW7 loss of function-linked epithelial malignancies through SOX9.

However, several key issues should be addressed to enhance the quality and impact of the paper. The following suggestions outline areas for improvement:

Major Critiques

1. FBW7 Protein Expression: The authors should demonstrate FBW7 expression in the kidney and validate FBXW7 deletion at the protein level in Cdh16-expressing nephron epithelia in FBXW7cKO animals vs. controls.
2. Figure 1F: To determine if the Ki67-based proliferation response is unique to FBW7-deficient nephron epithelia, provide analysis of Ki67+mG+ versus Ki67+mG- tubular epithelial cells in Cdh16Cre/+;ROSA26mT/mG;Fbxw7fl/fl kidneys.
3. Figure 2E, F: The histological images provided do not convincingly depict the pathology.
4. Figure 3D, E, F: Quantifying proteinuria using total protein (g/L) may be misleading. Instead, the authors should provide the urine protein-to-creatinine ratio at the specified time points. They should also explain the lower urinary pH in Fbxw7 cKO animals compared to controls, or vice versa, and clarify why control animals (Fbxw7 fl/fl) cannot acidify urine. Including biochemical profiles of age-matched wild-type controls could provide further insight into urinary pH impact.
5. Figure 4D: What are authors trying to show via Vimentin staining? Avoid vimentin as a marker of fibrosis.
6. Figure 4G: This image appears confusing. Based on the morphology, the mG+ tubular epithelia seem to show the medullary region, while the knockout appears to be from the cortical region. A thorough spatial and temporal characterization of the α SMA+ myofibroblast response in various regions, including nephron epithelia-cell-type and the cystic region, would be helpful. Additionally, replace mean fluorescence intensity (MFI) quantification with region-specific, percentage area (% area) showing the extent of area with α SMA+ myofibroblasts and F4/80+ macrophages.
1. Figure 6A: Characterize the SOX9 response in relation to nephron tubular epithelial cell type, apicobasal polarity, and proximal tubular brush border, in addition to cysts in Cdh16Cre/+;ROSA26mT/mG;Fbxw7fl/fl and controls. Temporal kinetics of SOX9 response in such kidneys would provide relationship with FBXW7 deficiency and progressive fibrosis. For example, P3 vs. 4 week vs. 8 weeks, in addition to the authors time-point. Validation of Cdh16Cre/+;Fbxw7fl/fl; Sox9fl/+ vs. corresponding controls with respect to SOX9 removal is crucial. To support impact on fibrosis, provide other fibrotic markers such as Col1a1 and Col3a1. In addition to SOX9 (already shown), provide Wnt4 levels in Fbxw7 KO cell line vs. control.
7. Quantification of α SMA+ Myofibroblasts (Figure 6J): Replace MFI quantification with % area.
8. Figure 6K: The images provided have excessive background and poor quality. Provide improved images and/or RNAScope based characterization of the Wnt4 response.

Minor Suggestions

1. Human Relevance: Since FBXW7 is linked to Wilms tumor and renal cell cancer in humans, the authors should discuss if their model exhibits features suggestive of these pathologies. If not, they should explain why the model reflects an NPHP-like disease in mice rather than these human pathologies.
2. Panel labeling in Figure 1F: The full label is missing. Specify the nuclear stain used, such as DAPI, instead of just "nuclei."
3. Figure 2A: Report the proportion of FBW7-deficient nephron epithelia showing apoptotic cells, and whether any nephron epithelial cell-type exhibits exaggerated apoptosis to FBXW7 loss. Here, the authors might utilize cleaved caspase-3 co-immunostainings.
4. Figures 5F, G: Specify the antibody used for cilia labeling rather than using the term "cilia."
5. In all relevant panels, replace "nuclei" with the specific stain or antibody used, such as DAPI.
6. Supplemental Figure 4: The pHH3 response is unconvincing. Include high-magnification images of EDU+ cells and their relationship to cystic epithelia.
7. Supplemental Figure 7A: cMYC staining lacks clarity. Provide a clearer image.

Referee #3 (Remarks for Author):

In their present study, Leonidas Tsiokas and collaborators have generated and characterized a new conditional mouse model where FBW7, the substrate recognition component of an evolutionary conserved SCF (complex of SKP1, CUL1 and F-box protein)-type ubiquitin ligase, was invalidated in kidney tubular epithelial cells. They provide strong evidence that this new mouse model presents with kidney defects similar to nephronophthisis, a renal ciliopathy, including polyuria and polydipsia linked to signs of interstitial fibrosis, thickening of the tubular basement membrane, tubular atrophy and cysts. They provide some mechanistic clues showing that TMEM237, a protein/gene linked to a cerebellar ciliopathy (Joubert syndrome), is downregulated in FBW7 KO epithelial cells and that SOX9, a direct target of FBW7 previously involved in fibrosis, is increased leading to increased WNT4 expression. This study provides interesting new results and the conclusions raised by the authors are globally sound. I however have several concerns which are listed below.

Major concerns:

- The claim that the described model is the first one to recapitulate juvenile NPHP should be revised. Several other orthologous and non-orthologous mouse models have been developed which show some similarities with the kidney defects found in humans (GLIS2/NPHP7, AATF, LKB1).
- The claim that kidney-specific FBW7 KO recapitulates NPHP is "over-stated". The kidney features presented here can be found in other nephropathies (diabetic nephropathy, lithium nephropathy). A clear link with human NPHP is lacking. What about FBW7 expression in kidney biopsies or cells coming from NPHP patients? Is there any evidence that SOX9/WNT4 is perturbed in NPHP?
- A more complete analysis of results obtained in IMCD3 cells should be provided. A single ciliary protein was picked up but the main information that could be obtained through a global analysis of these results is lacking. What about the 164 other dysregulated proteins?
- There is no link between the hypothesis-driven SOX9/WNT4 results and the ones obtained in the proteomic analysis of FBW7 KO IMCD3 cells. In addition, it is claimed that SOX9 acts through WNT4 but what about TMEM237 expression in SOX9 heterozygous mice? Is it rescued in these conditions? What is the mechanism linking the loss of FBW7, increased SOX9 expression and loss of TMEM237?
- A more global unbiased analysis of the FBW7 KO mice is also lacking. At least, RNAseq studies of the kidneys would help to better understand the pathways involved in this interesting new model and to allow to compare with the datasets available for other NPHP-related ones in which, to my knowledge, the SOX9/WNT4 axis was not yet involved.
- Some results are not quantified (basement membrane in Fig2E,F; ciliogenesis (% of ciliated cells and ciliary length) in Fig5F, ciliogenesis (% of ciliated cells) in Fig5.G,H)

Minor concerns:

- In the introduction, several points concerning NPHP are confusing. PKD is a specific kidney disease and most syndromic ciliopathies present with kidney defects not considered as PKD but as NPHP related. There are actually more than 25 genes that were involved in NPHP. It is said that there is no mouse model that "accurately mimic the early renal function decline seen in human NPHP", this is confusing because juvenile/late onsets represent the majority of NPHP cases.
- In the introduction, clarify the role/function of FBW7 more deeply.
- Follow the KDIGO recommendations: use "kidney" instead of "renal" when referring to kidney function and "kidney failure" instead of "ESKF/ESRF".
- Data showing the efficiency of the invalidation of FBW7, SOX9 and c-Myc at the protein level in kidneys are missing.
- In juvenile NPHP patients, the kidneys are either the same size or smaller. In the results it is said "without an increase in the 2KW/BW" but that is not what expected for a model of NPHP.
- Cre is expressed "at low level in proximal tubules and higher levels in cortical collecting ducts and cortical thick ascending limbs of Henle's loops" (Shao et al, JASN, 2002). Please rephrase in the results section "in the adult kidney, this promoter is active in

the ...", to specify that the Cre is in fact active at E10.5.

- Is supFig1A up-side down? If deletion, the indicated band should be smaller?
- The Sup Fig1B shows a very weak (if any) expression of the Cre in the proximal tubules as described by the team of Igarashi. Quantification should be provided. With such a low Cre expression in the proximal tubules, how explain so much LTL positive cysts (supFig2B)? Cyst incidence in the different nephron segments should be quantified (supFig2B-G).
- Positive controls for WB in Fig.S6 are missing.
- TMEM237 is supposed to be present at the transition zone (ciliary base) but stainings in Fig5D show a diffuse cytoplasmic staining in tubular cells. In addition, TMEM237 appeared to be expressed in some tubes only (Fig5D). Could the authors comment on these points (expression in which segments?) and provide additional control experiments for the used antibody?
- It is not clear if KI67, pH3 and EDU stainings were quantified in tubular cells or in all cells?
- Uncropped blots should be provided.
- To my knowledge acidosis is not classically reported in NPHP patients.
- Provide GEO with full data of proteomics.
- The image shown in Fig 5F is weird and does not look like a section of a cyst but more like a sheet of tubular cells.

- in the discussion:

"NPHP being the most common variants" > NPHP1

"we discovered that SOX9, but not c-MYC, serves as a crucial regulator of fibrosis in NPHP" > this remains to be directly investigated in NPHP conditions. It is clear that SOX9 plays a role in the presented FBW7 model but it remains an open question in the case in NPHP. What about SOX9 in ciliopathy conditions?

"These findings establish our Fbxw7-deletion-based mouse model as a valuable tool for studying NPHP pathology and lay the groundwork for developing therapeutic strategies that target SOX9-WNT4- dependent fibrosis in NPHP" > Again, the authors should provide evidence that this pathway is also altered in NPHP conditions (in vivo mouse models or in vitro mIMCD3 cells).

"a complete loss of FBW7 function. This contrasts with the hypomorphic mutations frequently observed in the reported NPHP-causing genes" > This is not correct, the most common cause of juvenile NPHP is a complete loss of the NPHP1 gene.

We thank the reviewers for their constructive feedback, thoughtful evaluation, and suggestions for improvement. We are pleased that the reviewers recognize the novelty and significance of our study, acknowledge the robust experimental design, rigorous statistical analyses, and the broader implications of our study for understanding the pathophysiology of kidney diseases. In the revised version, we have addressed the vast majority of the issues raised by adding a significant amount of new data and clarifications. A key addition is the evidence of increased SOX9 expression in a human NPHP patient biopsy (Figure 7A), reinforcing the translational relevance of our findings and the significance of our mouse model in studying the molecular mechanisms underlying NPHP-like conditions.

Referee .1 (Remarks for Author):

1. The Fbxw7-deficient NPHP-like model may be further used for studying renal function decline in other kidney diseases; in other words, findings could similarly apply to other kidney cystic diseases, such as ADPKD.

Our unpublished data show that in ADPKD mouse models, Fbxw7 mRNA and protein are increased, suggesting that the molecular bases of ADPKD and syndromic forms of PKD are different. This differential pattern of FBW7 expression in syndromic forms of PKD (NPHP-like pathology in this manuscript) versus ADPKD is also seen with GLIS2, whose loss-of-function mutations or deletion leads to NPHP, whereas its upregulation drives cyst formation in Pkd1 orthologous ADPKD mouse models (PMID: 38693102). Our data showing the upregulation of Fbxw7 mRNA and protein in ADPKD tissues are part of a separate manuscript and fall outside the scope of the current study. However, they are included in an earlier version of our manuscript, which is available in bioRxiv (PMID: 38464230). Here, we focus on the essential role of Fbxw7 in kidney homeostasis and preservation of renal function.

2. Although the SOX9-WNT4 axis may be in focus, other classic pro-fibrotic pathways associated with renal fibrosis, including that of TGF- β , are not pursued. This would, at best, diminish the comprehensive value of mechanistic insight and obscure potential pathway interplay that may fully explain NPHP fibrosis.

Please see Response Fig. 1 below. We assessed TGF- β pathway activation in our mouse model by pSMAD3 staining. In aged mice, pSMAD3 staining was evident in a small number of tubules, suggesting increased TGF- β signaling at later stages of disease progression. Based on these data, we have revised the discussion, suggesting the possible involvement of TGF- β signaling in augmenting fibrosis in our mouse model.

Response Fig. 1: Loss of FBW7 results in modest pSMAD3 activation in aged mice. (A-B) Representative images and quantification of pSMAD3 from 7-month-old kidneys of Fbxw7^{fl/fl} and Cdh16Cre;Fbxw7^{fl/fl} mice. (A) The white arrows show pSMAD+ cells in the cystic epithelium. Nuclei are stained with DAPI (blue). Scale bar: 20 μ m. (B) Each data point represents the percentage of pSMAD+ cells scored from cystic tubules per animal. Statistical analysis was performed using the Mann-Whitney test and is presented as the mean \pm SEM (: $p < 0.05$).*

3. The research will pursue renal function impairment by defects in urine concentration as well as through serum markers like BUN, creatinine, and cystatin C. Further work using GFR measurements or histological markers of glomerular injury would support a more advanced understanding of renal function loss in the context of FBXW7 deletion.

Please see Appendix Figure S3. We have added histological images highlighting glomerular damage, including glomerular cysts and Bowman's capsule thickening, reinforcing the structural manifestations of glomerular injury associated with the loss of FBW7. These new data together with blood work — BUN, creatinine, and cystatin C—shown in Fig. 1C-E, support the statement that Fbxw7 deletion leads to a decline in renal function.

4. Although the paper points out the increase in F4/80-positive macrophages as an indication of inflammation in kidneys with Fbxw7 deletion, the details are not stated.

We have added details to the discussion section.

5. Even in NPHP, it is regarded that inflammation is one of the most important factors in the development of renal fibrosis. Inflammatory responses can greatly contribute to the development of renal fibrosis, and further investigation into the link between inflammation and fibrosis may greatly enhance our understanding of the similar pathophysiology as seen in NPHP.

We do agree with the comment, and a future study dedicated to this connection, perhaps in a mouse model with a global deletion (or a pathogenic mutation) in one of the NPHP genes, may be more appropriate for such studies. In our study, we focused on the deletion of Fbxw7 in epithelial cells, and thus, the observed immune cell infiltration and inflammation are secondary to epithelial dysfunction. Addressing the link between inflammation and renal fibrosis in NPHP would require a different set of experiments, such as immune cell depletion in our model, to assess their specific contribution to fibrosis and the underlying molecular mechanisms. Nonetheless, we acknowledge the significance of this future direction and have revised the discussion.

6. In this study, findings are based on the mouse model alone, and how well these results will translate into human NPHP or other autosomal recessive cystic kidney disorders is not known. Human tissue validation or genetic studies in NPHP patients could strengthen translational applicability.

Please see Figure 7A-B. We have included data from the biopsy of an NPHP patient carrying a mutation in the NPHP1 gene, representing the most common form of NPHP. Representative images from three different regions of the biopsy show an increase in SOX9 expression in the kidney tubules compared to controls. Additionally, we have included data from the CaggCre^{ERT2};Ift88^{off} mouse model, which features blunted cilia, a condition seen in many NPHP-causing gene mutations. This mouse model also shows a significant increase in SOX9 expression in cystic compared to control tubules (Figure 7C-D). Together, these findings enhance the translational relevance of our study and illustrate how the Fbxw7-deletion-based NPHP-like mouse model offers important insights into understanding the molecular mechanism(s) underlying NPHP.

Referee #2 (Remarks for Author):

This study addresses two major issues in nephronophthisis (NPHP) cystic kidney disease: (1) generating a mouse model that replicates key features of human NPHP, including progressive renal dysfunction, and (2) identifying mechanisms driving fibrosis in genetic cystic kidney disease, which are poorly understood. The study successfully generated an NPHP-like disease model by deleting FBW7 in Cdh16-expressing nephron epithelia. This led to a SOX9 response, an FBW7-regulated gene, in FBW7-deficient Cdh16+ epithelia, resulting in fibrosis via Wnt4, a short-range secretory ligand. Importantly, partial SOX9 deletion in Cdh16-specific, FBW7-deficient animals reduced fibrosis and promoted renal function recovery, as indicated by significantly lower serum creatinine and cystatin C levels. With respect to NPHP-like disease, an unbiased identification of TMEM237—a gene associated with Joubert syndrome-related disorders (JSRD), which includes cystic kidney disease as one of the manifestations, provide

additional supports to the development of a cystic kidney disease-like phenotype in this FBW7-deficient mouse model. One of the hallmarks of kidney disease in JSRD patients are corticomedullary cystic kidneys with exaggerated fibrosis.

Taken together, this study represents a significant advancement in the field. Further, this study is suitable for broad readership because ciliopathies involve different organs, and these findings might have broad implications, for example, in pathological remodeling of tumor microenvironment in FBXW7 loss of function-linked epithelial malignancies through SOX9.

However, several key issues should be addressed to enhance the quality and impact of the paper. The following suggestions outline areas for improvement:

Major Critiques

1. FBW7 Protein Expression: The authors should demonstrate FBW7 expression in the kidney and validate FBXW7 deletion at the protein level in Cdh16-expressing nephron epithelia in FBXW7cKO animals vs. controls.

Please see Appendix Figure S1C-D. We have included immunohistochemical data showing FBW7 levels in Cdh16Cre;Fbxw7^{fl/fl} kidneys compared to controls. It should be noted though, that detecting FBW7 at the protein level in tissue sections using indirect immunofluorescence or immunohistochemistry is not trivial. We have tried 7 different antibodies, and one (Bethyl A301-720A) showed a significant reduction of expression in Cdh16Cre;Fbxw7^{fl/fl} kidneys.

2. Figure 1F: To determine if the Ki67-based proliferation response is unique to FBW7-deficient nephron epithelia, provide analysis of Ki67+mG+ versus Ki67+mG- tubular epithelial cells in Cdh16Cre/+;ROSA26mT/mG;Fbxw7fl/fl kidneys.

Please see Figure 1F-G. We have provided mG+Ki67+ versus mG-Ki67+ cells representative images and quantification.

3. Figure 2E, F: The histological images provided do not convincingly depict the pathology.

Please see Figure 2E-H. We have revised the images and provided quantification.

4. Figure 3D, E, F: Quantifying proteinuria using total protein (g/L) may be misleading. Instead, the authors should provide the urine protein-to-creatinine ratio at the specified time points. They should also explain the lower urinary pH in Fbxw7 cKO animals compared to controls, or vice versa, and clarify why control animals (Fbxw7 fl/fl) cannot acidify urine. Including biochemical profiles of age-matched wild-type controls could provide further insight into urinary pH impact.

Please see Figure 3D, E, and F. We revised the graphs representing the protein-to-creatinine ratio. The biochemical profile in serum samples revealed a reduction in tCO₂ levels, indicative of metabolic acidosis (Appendix Figure S6). This finding offers an explanation of urine acidification, as metabolic acidosis often results in lower urine pH by stimulating increased hydrogen ion excretion by the kidneys. Such acid-base dysregulation is commonly observed in chronic kidney diseases and Joubert syndrome patients displaying NPHP-like pathology (PMID: 35003283 and 32944439), where impaired acid excretion leads to systemic and urinary acidification.

5. Figure 4D: What are authors trying to show via Vimentin staining? Avoid vimentin as a marker of fibrosis.

As suggested, we have removed vimentin data and added COL1A1 data (Figure 4E-F and Figure 6G-H).

6. Figure 4G: This image appears confusing. Based on the morphology, the mG+ tubular epithelia seem to show the medullary region, while the knockout appears to be from the cortical region. A thorough

spatial and temporal characterization of the α SMA+ myofibroblast response in various regions, including nephron epithelia-cell-type and the cystic region, would be helpful. Additionally, replace mean fluorescence intensity (MFI) quantification with region-specific, percentage area (% area) showing the extent of area with α SMA+ myofibroblasts and F4/80+ macrophages.

1. Figure 6A: Characterize the SOX9 response in relation to nephron tubular epithelial cell type, apicobasal polarity, and proximal tubular brush border, in addition to cysts in Cdh16Cre/+;ROSA26mT/mG;Fbxw7fl/fl and controls. Temporal kinetics of SOX9 response in such kidneys would provide relationship with FBXW7 deficiency and progressive fibrosis. For example, P3 vs. 4 week vs. 8 weeks, in addition to the authors timepoint. Validation of Cdh16Cre/+;Fbxw7fl/fl; Sox9fl/+ vs. corresponding controls with respect to SOX9 removal is crucial. To support impact on fibrosis, provide other fibrotic markers such as Col1a1 and Col3a1. In addition to SOX9 (already shown), provide Wnt4 levels in Fbxw7 KO cell line vs. control.

We have revised the α SMA images in Figure 4G to specifically represent the kidney cortex region, confirmed by co-staining with LTA, a proximal tubular marker. We have provided a comprehensive spatial and temporal analysis of SOX9+ cells and α SMA+ percent area across different kidney regions (Appendix Figure S10 and S11). Our analysis revealed that, unlike acute kidney injury models, our model exhibited a gradual increase in SOX9+ cells over time, which correlated with the α SMA+ myofibroblast response. To ensure proper representation of age-matched cohorts, we included data from an additional P16 time point alongside the 3- and 7-month time points. These analyses account for the α SMA+ myofibroblast response with respect to Sox9 removal. Notably, SOX9+ cells were predominantly localized to the cortico-medullary border and cortex, with minimal SOX9+ cells in the medullary region, which only appeared convincingly in aged mice (Figure 6A, Appendix Figure S10). Consecutively, upon Sox9 removal, amelioration of α SMA+ myofibroblast response was predominantly seen in the cortex region as opposed to modest effects in the medullary region (Appendix Figure S11). We also characterized the SOX9 distribution in both cystic and non-cystic cells in the cortico-medullary and cortex regions (Figure 6A and Appendix Figure S10A-C). We have also provided SOX9 distribution across proximal tubules with brushed borders as well as in cells where apicobasal polarity is lost (Appendix Figure S10E-H). For F4/80+ macrophages, we replaced MFI with cortex-specific percentage area (Figure 4B). We have also included COL1A1 staining (Figure 4E-F and Figure 6G-H). Additionally, qPCR data from the IMCD3 Fbxw7-null cell line revealed significantly elevated Wnt4 transcript levels compared to controls (Figure 6M). We have revised the text in the “Results” and “Discussion” sections, which elaborate on all of the above findings in relevant places.

7. Quantification of α SMA+ Myofibroblasts (Figure 6J): Replace MFI quantification with % area. %area of α SMA graph

Please see Figure 6J. We have revised the graphs to show quantification based on the % area of α SMA staining.

8. Figure 6K: The images provided have excessive background and poor quality. Provide improved images and/or RNAScope based characterization of the Wnt4 response.

Please see Figure 6K. We have revised the images.

Minor Suggestions

1. Human Relevance: Since FBXW7 is linked to Wilms tumor and renal cell cancer in humans, the authors should discuss if their model exhibits feature suggestive of these pathologies. If not, they should explain why the model reflects an NPHP-like disease in mice rather than these human pathologies.

While FBXW7 mutations are associated with Wilm’s tumor and renal cell carcinoma (RCC) in humans, our mouse model does not exhibit spontaneous tumor formation up to 10 months of age. In mouse

models, deletion of well-established tumor suppressors such as Vhl (also an E3 ligase) does not independently lead to spontaneous tumor formation; it requires additional driver mutations in p53 or Rb1 (PMID: 25023703 and 28553932). This suggests that FBW7 loss alone may not be sufficient to initiate tumorigenesis. Interestingly, genes associated with syndromic forms of polycystic kidney disease, such as those linked to NPHP, are involved in DNA damage response pathways—a function most relevant in cancer settings, which is also shared by FBW7, as a tumor suppressor (PMID: 22863007 and 34760892). Additionally, disruption of either FBW7 or NPHP-associated genes leads to cilia-related defects. Given these functionally overlapping relationships, it is not entirely surprising that FBW7 deletion results in an NPHP-like pathology in the kidney. An important avenue for future research is to determine whether NPHP-like pathology predisposes the kidney to tumor development if a secondary genetic hit occurs. This raises the intriguing possibility of a link between ciliopathy-associated kidney diseases and pre-malignant states.

2. Panel labeling in Figure 1F: The full label is missing. Specify the nuclear stain used, such as DAPI, instead of just "nuclei."

Labeling has been revised at respective places.

3. Figure 2A: Report the proportion of FBW7-deficient nephron epithelia showing apoptotic cells, and whether any nephron epithelial cell-type exhibits exaggerated apoptosis to FBXW7 loss. Here, the authors might utilize cleaved caspase-3 co-immunostainings.

Please see Figure 2B and D. We have revised the quantification of tubular cell death, which is now represented by the percentage of mG+ cells that are TUNEL-positive. While TUNEL-positive cells were observed throughout the kidney, they were predominant in the medullary segments.

4. Figures 5F, G: Specify the antibody used for cilia labeling rather than using the term "cilia."
Add replace with acty-tubulin at all places.

Labeling has been revised at respective places.

5. In all relevant panels, replace "nuclei" with the specific stain or antibody used, such as DAPI.
Add replace with DAPI at all places.

Labeling has been revised at respective places.

6. Supplemental Figure 4: The pHH3 response is unconvincing. Include high-magnification images of EDU+ cells and their relationship to cystic epithelia.

Please see Appendix Figure S5B and D. High-magnification images have been provided for both EDU and pHH3 staining on ROSA^{mT/mG} background to pinpoint the proliferating epithelial cells.

7. Supplemental Figure 7A: cMYC staining lacks clarity. Provide a clearer image.

Please see Appendix Figure S12A. We have revised the respective images.

Referee #3 (Remarks for Author):

In their present study, Leonidas Tsiokas and collaborators have generated and characterized a new conditional mouse model where FBW7, the substrate recognition component of an evolutionary conserved SCF (complex of SKP1, CUL1 and F-box protein)-type ubiquitin ligase, was invalidated in kidney tubular epithelial cells. They provide strong evidence that this new mouse model presents with kidney defects similar to nephronophthisis, a renal ciliopathy, including polyuria and polydipsia linked to sign of interstitial fibrosis, thickening of the tubular basement membrane, tubular atrophy and cysts.

They provide some mechanistic clues showing that TMEM237, a protein/gene linked to a cerebellar ciliopathy (Joubert syndrome), is downregulated in FBW7 KO epithelial cells and that SOX9, a direct target of FBW7 previously involved in fibrosis, is increased leading to increase WNT4 expression. This study provide interesting new results and the conclusions raised by the authors are globally sounded. I however have several concerns which are listed below.

Major concerns:

1. The claim that the described model is the first one to recapitulate juvenile NPHP should be revised. Several other orthologous and non-orthologous mouse models have been developed which show some similarities with the kidney defects found in humans (GLIS2/NPHP7, AATF, LKB1).

We thank the reviewer for bringing these studies to our attention. We have revised the text accordingly.

2. The claim that kidney-specific FBW7 KO recapitulates NPHP is "over-stated". The kidney features presented here can be found in other nephropathies (diabetic nephropathy, lithium nephropathy). A clear link with human NPHP is lacking. What about FBW7 expression in kidney biopsies or cells coming from NPHP patients? Is there any evidence that SOX9/WNT4 is perturbed in NPHP?

Please see Fig. 7A and B, as well as our response to Reviewer #1, Point #6.

3. A more complete analysis of results obtained in IMCD3 cells should be provided. A single ciliary protein was picked up but the main information that could be obtained through a global analysis of these results is lacking. What about the 164 other dysregulated proteins?

We have included a comprehensive pathway analysis from the proteomic screen of IMCD3 cells (Appendix Figure S9). Across datasets, the majority of upregulated pathways were associated with metabolic processes, particularly fatty acid oxidation, which FBW7 is known to regulate by influencing fatty acid synthesis and transport, albeit in a cancer setting. In contrast, most downregulated pathways were related to proteasomal degradation and ubiquitylation, likely representing a compensatory response to the loss of FBW7, a substrate recognition component of the SCF-type ubiquitin ligase responsible for identifying proteins for proteasomal degradation. While these pathway analyses provide global insights into the effects of FBW7 loss, they were less informative in identifying specific molecular players directly linked to the NPHP-like pathology observed in the Fbxw7-null kidneys. We focused on TMEM237 because it is known to affect cilia structure similarly to FBW7 and is mutated in Joubert patients with a renal pathology closely related to NPHP. No other proteins in the top 20 lists stood out as direct candidates implicated in NPHP-like pathology in humans (Dataset EV1). Thus, we decided to include the data showing TMEM237 downregulation in our manuscript because of its potential functional implications in the NPHP-like phenotype seen in our mouse model and Joubert patients. While other dysregulated proteins may work together or independently of TME237 to modulate cystogenesis, apoptosis, and inflammation, further in vivo genetic studies are necessary to determine their physiological relevance and lay the groundwork for future investigations.

4. There is no link between the hypothesis-driven SOX9/WNT4 results and the ones obtained in the proteomic analysis of FBW7 KO IMCD3 cells. In addition, it is claimed that SOX9 acts through WNT4 but what about TMEM237 expression in SOX9 heterozygous mice? Is it rescued in these conditions? What is the mechanism linking the loss of FBW7, increased SOX9 expression and loss of TMEM237?

We did not see a significant rescue of TMEM237 expression upon Sox9 removal in our mouse model (Appendix Figure S14A-B). Additionally, SOX9 overexpression experiments were also performed to investigate if SOX9 overexpression altered TMEM237 expression in HEK293T cells, independent of any

changes in FBW7. Analysis of TMEM237 mRNA and protein levels revealed no significant effect of SOX9 overexpression on TMEM237 expression (Appendix Figure S14C-D). Based on these observations, it is likely that FBW7 target proteins other than SOX9 might be involved in negatively regulating TMEM237 levels in IMCD3 cells and possibly in the mouse kidney. As we stated in the manuscript, the downregulation of TMEM237 in Fbxw7 mutant kidneys is an indirect effect of the deletion of Fbxw7.

5. A more global unbiased analysis of the FBW7 KO mice is also lacking. At least, RNAseq studies of the kidneys would help to better understand the pathways involved in this interesting new model and to allow to compare with the datasets available for other NPHP-related ones in which, to my knowledge, the SOX9/WNT4 axis was not yet involved.

Although an RNAseq analysis could provide a global view of gene expression changes in single and double mutant mouse kidneys, it would represent indirect effects of these deletions, especially for FBW7, which is a component of an E3 ligase. Instead, we performed a more controlled analysis in mIMCD3 cells that resulted in the identification of direct targets of FBW7 and gave us a global view of proteome changes and associated pathways in cell culture. A more in-depth analysis using a combination of RNAseq and proteomics in Cdh16-positive cells isolated from our mouse model could be the subject of a future study.

6. Some results are not quantified (basement membrane in Fig2E,F; ciliogenesis (% of ciliated cells and ciliary length) in Fig5F, ciliogenesis (% of ciliated cells) in Fig5.G,H)

Respective figures are revised with quantifications.

Minor concerns:

1. In the introduction, several points concerning NPHP are confusing. PKD is a specific kidney disease and most syndromic ciliopathies present with kidney defects not considered as PKD but as NPHP related. There are actually more than 25 genes that were involved in NPHP. It is said that there is no mouse model that "accurately mimic the early renal function decline seen in human NPHP", this is confusing because juvenile/late onsets represent the majority of NPHP cases.

We have revised the text.

2. In the introduction, clarify the role/function of FBW7 more deeply.

We added more details about the function of FBW7 in the Introduction section.

3. Follow the KDIGO recommendations: use "kidney" instead of "renal" when referring to kidney function and "kidney failure" instead of "ESKF/ESRF".

Recommended text changes are made throughout the manuscript.

4. Data showing the efficiency of the invalidation of FBW7, SOX9 and c-Myc at the protein level in kidneys are missing.

For FBW7 invalidation, please refer to Reviewer #2, Point #1. Since SOX9 and c-MYC are known to have negligible expression in control adult kidney tissues, we assessed their invalidation by analyzing Cdh16Cre;Fbxw7^{ff} versus Cdh16Cre;Fbxw7^{ff};Sox9^{+/ff} or Cdh16Cre;Fbxw7^{ff};c-Myc^{+/ff} kidneys. These data showed a considerable reduction in SOX9+ and c-MYC+ nuclei in the respective mouse lines (Appendix Figure S1E-H). The residual SOX9 or c-MYC signals likely arise from the remaining allele of Sox9 or c-Myc. Additionally, it is worth noting that Fbxw7^{ff}, Sox9^{ff}, c-Myc^{ff}, and Cdh16Cre mouse lines, although not in the combination reported in this manuscript, have been extensively used and inactivation of these genes have been validated in previous studies.

5. In juvenile NPHP patients, the kidneys are either the same size or smaller. In the results it is said "without an increase in the 2KW/BW" but that is not what expected for a model of NPHP.

We have rephrased the sentence to enhance clarity.

6. Cre is expressed "at low level in proximal tubules and higher levels in cortical collecting ducts and cortical thick ascending limbs of Henle's loops" (Shao et al, JASN, 2002). Please rephrase in the results section "in the adult kidney, this promoter is active in the ...", to specify that the Cre is in fact active at E10.5.

We have revised the text.

7. Is supFig1A up-side down? If deletion, the indicated band should be smaller?

FBW7 deletion is genetically confirmed through PCR analysis by the presence of the upper band (~660 bp) (Appendix Figure S1B). The PCR primers and conditions were adopted from the study that initially characterized Fbxw7 deletion using the Fbxw7^{ff} mouse line, although with different Cre and tissue types (PMID: 18474632). The lower band likely corresponds to a non-specific product consistent across all PCR conditions and genotypes we tried.

8. The Sup Fig1B shows a very weak (if any) expression of the Cre in the proximal tubules as described by the team of Igarashi. Quantification should be provided. With such a low Cre expression in the proximal tubules, how explain so much LTL positive cysts (supFig2B)? Cyst incidence in the different nephron segments should be quantified (supFig2B-G).

We have included quantification and images of the distribution of cystic tubules across different kidney segments at both 3- and 7-month time points and images that show cystic mG+ tubules are also LTA-positive (Appendix Figure S2B-J). Although transgene expression in the proximal tubules is minimal using this driver, the possibility that Cre expression at levels sufficient to achieve recombination in late time points such as 7 months cannot be excluded. The Cdh16Cre is predominantly active in the medulla as opposed to the cortex (Appendix Figure S1A). Despite this, our data show that cyst formation mainly occurs in the corticomedullary border and cortex, including proximal tubules, while the medullary segments are less affected in the context of cysts. This pattern is clearly visible in kidney scans, where dilated/cystic tubules are most prominent in the corticomedullary border and cortex, with fewer cysts observed in the medulla (Figure 1A).

9. Positive controls for WB in Fig.S6 are missing.

Parental and empty-vector transfected cells are positive controls showing endogenous FBW7 levels in IMCD3 cells (Appendix Figure S8). Immunoprecipitations followed by Western blotting using two separate antibodies (one for immunoprecipitation and another for detection) identified the FBW7 band in two control samples and was deleted in knockout clones.

10. TMEM237 is supposed to be present at the transition zone (ciliary base) but stainings in Fig5D show a diffuse cytoplasmic staining in tubular cells. In addition, TMEM237 appeared to be expressed in some tubes only (Fig5D). Could the authors comment on these points (expression in which segments?) and provide additional control experiments for the used antibody?

We performed a Western blot using HEK293T cells overexpressing mouse TMEM237 as a control experiment to validate the commercial antibody used in our study. The antibody successfully detected mouse TMEM237, confirming its specificity (Appendix Figure S14E). In adult kidney tissue, TMEM237 expression was predominantly observed in the corticomedullary region, and notably, the loss of FBW7 affected TMEM237 expression exclusively in cystic tubules. To ensure proper comparison, we analyzed

cystic tubules where Cdh16Cre was partially active using a ROSA^{mT/mG} background. This confirmed that the representative tubule in the image inherently expressed TMEM237, while cells in regions where Fbxw7 was deleted (mG+) showed reduced TMEM237 levels, whereas those where Cdh16Cre was inactive (mT+) retained TMEM237 expression (Figure 5D). In some cysts, TMEM237 expression was nearly undetectable in almost all cystic cells, with some cells having faint residual signal, suggesting that its downregulation is closely linked to cystogenesis, particularly in the corticomedullary region (Appendix Figure S14A). Detection of the TMEM237 signal in the ciliary transition zone in tissue sections is technically challenging as the transition zone represents a tiny (~200 nm-wide) subcellular compartment. Our revised images (Figure 5D) clearly show TMEM237 co-localized with mT+ or mG+ signal at the plasma membrane, aligning with previous observations from another study that used a different TMEM237 antibody in a 100% confluent cell culture (Figure S6 from PMID: 22152675). To our knowledge, demonstration of TMEM237 presence in the transition zone in tissue sections has never been shown.

11. It is not clear if Ki67, pH3 and EDU stainings were quantified in tubular cells or in all cells?
We have revised the text to clarify that the quantifications were obtained from tubular cells. For Ki67, we have also included the quantification for non-tubular cells (Figure 1F-G).

12. Uncropped blots should be provided.
We have provided uncropped blots.

13. To my knowledge acidosis is not classically reported in NPHP patients.
Joubert syndrome patients experiencing kidney function decline have been reported to develop metabolic acidosis, which aligns with the downregulation of TMEM237 observed in our mouse model (PMID: 32944439).

14. Provide GEO with full data of proteomics.
We have included complete pathways analysis using different datasets (Appendix Figure S9) and provided IMCD3 proteomics data (Dataset EV1).

15. The image shown in Fig 5F is weird and does not look like a section of a cyst but more like a sheet of tubular cells.
We have revised the images.

16. in the discussion:
"NPHP being the most common variants" > NPHP1
We have revised the text.

"we discovered that SOX9, but not c-MYC, serves as a crucial regulator of fibrosis in NPHP" > this remains to be directly investigated in NPHP conditions. It is clear that SOX9 plays a role in the presented FBW7 model but it remains an open question in the case in NPHP. What about SOX9 in ciliopathy conditions?
Please refer to Reviewer #1, Point #6.

"These findings establish our Fbxw7-deletion-based mouse model as a valuable tool for studying NPHP pathology and lay the groundwork for developing therapeutic strategies that target SOX9-WNT4-dependent fibrosis in NPHP" > Again, the authors should provide evidence that this pathway is also altered in NPHP conditions (in vivo mouse models or in vitro mIMCD3 cells).

Please refer to Reviewer #1, Point #6.

"a complete loss of FBW7 function. This contrasts with the hypomorphic mutations frequently observed in the reported NPHP-causing genes" > This is not correct, the most common cause of juvenile NPHP is a complete loss of the NPHP1 gene.

We have revised the text.

11th Mar 2025

Dear Prof. Tsiokas,

Thank you for submitting your revised study. We have now received the reports from the referees. As you will see from the reports below, they are satisfied with the revisions, and I will therefore be able to accept your manuscript once the following minor issues are addressed:

1/ Referees' comments:

- Please address the remaining concerns/comments from referees #1 and #2 by adequate discussion/clarifications.

2/ Manuscript text:

- Please remove the yellow highlighted font and only keep in track changes mode any new modification.

- Please provide up to 5 keywords.

- "Materials and Methods" should be renamed "Methods":

o Mouse models: please detail housing and husbandry conditions.

o Humans samples: please include a statement confirming that the experiments conformed to the principles set out in the WMA Declaration of Helsinki and the Department of Health and Human Services Belmont Report.

- Please add a "Graphics" section to the "Methods" section following this format:

"Graphics:

(some of the... OR Figure #... OR synopsis) Graphics were created with BioRender.com."

- Data availability: Primary datasets produced in this study must be deposited in an appropriate public database before acceptance of the manuscript, and the accession numbers and database listed under 'Data Availability'. Please remove "Differential analysis of the proteomics data used in the study is provided as Dataset EV1. Raw files will be uploaded to a repository in the future.", and only list in this section the link to deposited datasets.

- Acknowledgements: the funding information provided in the manuscript and the submission system should match. Please check whether K01DK119375 and R01DK129255 should be added to the list of funders in our system.

- Please rename "Conflict of interest statement" to "Disclosure and competing interests" and place it after the Acknowledgements.

3/ Figures and Appendix:

- Given the number of figures currently in your Appendix, we would suggest making some of these supplementary figures 'Expanded View (EV) Figures' that are collapsible/expandable online. EV Figures should be cited as 'Figure EV1, Figure EV2' etc... in the text and their respective legends should be included in the main text after the legends of regular figures.

- For the figures that you do NOT wish to display as Expanded View figures, they should be bundled together with their legends in a single PDF file called *Appendix*, which should start with a short Table of Content including page numbers.

- The uncropped blots and gels should be removed from the Appendix and uploaded as source data files.

- EV Dataset: the dataset title should be added together with a short description of the dataset in a separate tab/worksheet.

- Please make sure that all figures/figure panels are referenced in the manuscript text in sequential order (currently, a callout is missing for Table EV1, and Appendix figures are not all called out sequentially).

- Figure re-use should be indicated in the figure legends (i.e. Figure 4G and Appendix Figure S11A). We also note a possible partial re-use within Appendix Fig. S11C - 3 months, please check and clarify.

- Please address the queries from our copy editors:

1. Please note that the exact p values are not provided in the legends of figures 1C, D, E, G; 2B, D, F, H; 3A-C, E-O; 4B, D, F, H; 5C, E, G, H, J, K; 6B, D, F, H, J, L, M-Q; 7D

2. Please indicate the statistical test used for data analysis in the legends of figures 5A

3. Please note that information related to n is missing in the legends of figures 1B, C, D, E; 2B, D, F, H; 3A-O; 4B, D, F, H; 5A, C, E, G, H; 6B, D, F, H, J, L, M-Q; 7D

4/ Source Data:

The files need to be re-organized and uploaded zipped as one file per figure. The readme files should be zipped to the corresponding figures.

5/ Checklist:

- Please fill in the subsection on inclusion/exclusion criteria.

- Please fill in the right column for Data Availability/primary datasets

6/ Thank you for providing The paper explained. Please add it to the manuscript main text file.

7/ Synopsis:

Thank you for providing a visual abstract. I have cropped a small portion to serve as a thumbnail on our website (attached).

Please let me know if you agree, or provide an alternative image (115 x 70 pixels).

8/ As part of the EMBO Publications transparent editorial process initiative (see our Editorial at <http://embomolmed.embopress.org/content/2/9/329>), EMBO Molecular Medicine will publish online a Review Process File (RPF) to accompany accepted manuscripts.

This file will be published in conjunction with your paper and will include the anonymous referee reports, your point-by-point response and all pertinent correspondence relating to the manuscript. Let us know whether you agree with the publication of the RPF and as here, if you want to remove or not any figures from it prior to publication.

I look forward to receiving your revised manuscript.

Yours sincerely,

Lise Roth

***** Reviewer's comments *****

Referee #1 (Remarks for Author):

After reviewing the revised manuscript and the authors' point-by-point responses, I find that the manuscript, "SOX9-dependent fibrosis drives renal function in nephronophthisis," has been significantly improved. The authors have addressed the vast majority of concerns raised by the reviewers through the addition of new data, methodological clarifications, and textual revisions, enhancing the study's scientific rigor and translational relevance. Below, I outline the key improvements and note a few areas that could still benefit from minor refinement.

Key Improvements:

- 1- The inclusion of SOX9 expression data from a human NPHP1 patient biopsy (Fig. 7A-B) is a major strength, significantly enhancing the translational value of the study. The additional data from the CaggCreERT2;lft88f/f mouse model (Fig. 7C-D), which mimics ciliary defects common in NPHP, further supports the relevance of the findings to human disease.
- 2- The addition of pSMAD3 staining data in aged mice (Response Fig. 1) responds to critics about exploring other pro-fibrotic pathways beyond SOX9-WNT4. This new evidence supports a potential secondary role for TGF- β signaling in later-stage fibrosis, enriching the mechanistic insights.
- 3- The authors have enhanced methodological transparency by quantifying TUNEL staining (Fig. 2B, D), replacing total protein with protein-to-creatinine ratios for proteinuria (Fig. 3D-F), and providing spatial-temporal characterization of α SMA and SOX9 responses (Appendix Figs. S10-S11). These revisions address concerns about data robustness and specificity.
- 4- The study now more convincingly demonstrates the SOX9-WNT4 axis as a driver of fibrosis and kidney function decline, supported by improved imaging (e.g., Fig. 6K) and the replacement of vimentin with COL1A1 staining (Fig. 4E-F, 6G-H), a more specific fibrosis marker.
- 5- The Fbxw7 knockout mouse model is thoroughly characterized through updated histological images (e.g., Fig. 2E-H, Appendix Fig. S3), quantifications of cyst progression, tubular degeneration, inflammation, and fibrosis (e.g., Figs. 1-4), and serum markers of kidney function decline (Fig. 1C-E), aligning well with NPHP-like pathology.

Areas for Further Consideration:

1- The authors provide new data showing no rescue of TMEM237 expression in Sox9 heterozygous mice (Appendix Fig. S14A-B) and no effect of SOX9 overexpression on TMEM237 (Appendix Fig. S14C-D). While this clarifies that TMEM237 downregulation is independent of SOX9, the mechanism linking FBW7 loss to TMEM237 remains unresolved, as acknowledged in the manuscript. Essentially, the authors have ruled out SOX9 as a direct mediator of TMEM237 downregulation but haven't yet pinpointed the exact mechanism responsible for this effect

2- The study robustly links SOX9-WNT4 to fibrosis, but the modest pSMAD3 activation suggests other pathways (e.g., TGF- β) may contribute independently, particularly in non-SOX9 regions like the medulla. This is noted but not fully explored.

The authors have made an enough effort to address the reviewers' concerns, resulting in a substantially strengthened manuscript. The addition of human patient data, improved methodological rigor, and enhanced model characterization elevate its scientific quality and relevance. The remaining areas-namely, the unresolved FBW7-TMEM237 mechanism represent minor limitations that do not detract from the study's core findings. I recommend acceptance.

Referee #2 (Comments on Novelty/Model System for Author):

The authors have established a clinically meaningful model of NPHP-like kidney disease and addressed an important, poorly understood question in the field of cystic kidney disease: What is the molecular driver of fibrosis-the pathology that leads such kidneys to end-stage kidney disease?

The herein identified pathways might be applicable to other epithelia-based organ system that are involved in cystic diseases.

Referee #2 (Remarks for Author):

This reviewer is overall satisfied by the authors responses.

However, three clinical statements/findings needs to corrected/reexamined:

Page 6: Heterozygous Cdh16Cre;Fbxw7+/f mice exhibited no significant changes in cyst development, 2KW/BW, or kidney function at 3 months of age (Appendix Figure S4). However, by 7 months, these mice experienced a mild decline in kidney function, indicated by elevated BUN levels (Appendix Figure S4C).

COMMENT: In this setting of significantly increased urine output, a solitary increase in BUN levels is more likely to reflect relative volume depletion rather than a mild decline in kidney function. Therefore, please avoid making this claim.

2. Page 7: In these mice, a significant reduction in urine specific gravity and total protein-creatinine ratio was observed, especially in aged mice, which is indicative of a compromised ability to concentrate urine (Figure 3A-F). This defect was primarily driven by an increase in urine output, which correlated with elevated water consumption (Figure 3G-L).

COMMENT: Please note that a significant reduction in the urine protein-to-creatinine ratio is not indicative of a compromised ability to concentrate urine. Therefore, this statement is incorrect.

Increasing proteinuria, as quantified by the urine protein-to-creatinine ratio, is a strong and independent predictor of chronic kidney disease (CKD) progression, regardless of the underlying etiology. Therapies that significantly reduce proteinuria retard the progression of CKD.

Therefore, in this setting of CKD secondary to NPHP-like kidney disease, as evidenced by overall biochemical and histopathological findings, this paradoxical result raises questions about the methodology used to assay urine spot creatinine and protein. The authors should critically examine the methodology used to calculate the spot urine protein-to-creatinine ratio in this animal study and determine whether a true difference in proteinuria exists.

3. The acidic urine pH (Figure 3M-O) and low serum tCO₂ levels in aged mice (Appendix Figure S6) further suggested symptoms of acidosis resulting from altered renal handling of electrolytes and solutes.

COMMENT: Suggest to remove word "symptoms". These are findings and not "symptoms". Please correct the statement after removing the word symptom.

Referee #3 (Remarks for Author):

Suitable for publication.

We are pleased that the reviewers found the revisions satisfactory and support the manuscript's acceptance. We have addressed the minor concerns in this revision and updated the text and figures accordingly.

1. Referees' comments:

Referee #1 Areas for Further Consideration:

1- The authors provide new data showing no rescue of TMEM237 expression in Sox9 heterozygous mice (Appendix Fig. S14A-B) and no effect of SOX9 overexpression on TMEM237 (Appendix Fig. S14C-D). While this clarifies that TMEM237 downregulation is independent of SOX9, the mechanism linking FBW7 loss to TMEM237 remains unresolved, as acknowledged in the manuscript. Essentially, the authors have ruled out SOX9 as a direct mediator of TMEM237 downregulation but haven't yet pinpointed the exact mechanism responsible for this effect

Please see lines 14-18, page 14. We have acknowledged this limitation in the discussion section.

2- The study robustly links SOX9-WNT4 to fibrosis, but the modest pSMAD3 activation suggests other pathways (e.g., TGF- β) may contribute independently, particularly in non-SOX9 regions like the medulla. This is noted but not fully explored.

Please see lines 21-24, page 15. We have acknowledged these possibilities in the discussion section. Fully exploring the contribution of other pathways in the medulla, such as TGFbeta, would require the generation of genetic mouse models and is beyond the scope and focus of our present study.

Referee #2 (Remarks for Author):

This reviewer is overall satisfied by the authors responses.

However, three clinical statements/findings needs to corrected/reexamined:

Page 6: Heterozygous Cdh16Cre;Fbxw7+/f mice exhibited no significant changes in cyst development, 2KW/BW, or kidney function at 3 months of age (Appendix Figure S4). However, by 7 months, these mice experienced a mild decline in kidney function, indicated by elevated BUN levels (Appendix Figure S4C). COMMENT: In this setting of significantly increased urine output, a solitary increase in BUN levels is more likely to reflect relative volume depletion rather than a mild decline in kidney function. Therefore, please avoid making this claim.

Please see lines 18-20, page 7. We have revised the text.

2. Page 7: In these mice, a significant reduction in urine specific gravity and total protein-creatinine ratio was observed, especially in aged mice, which is indicative of a compromised ability to concentrate urine (Figure 3A-F). This defect was primarily driven by an increase in urine output, which correlated with elevated water consumption (Figure 3G-L). COMMENT: Please note that a significant reduction in the urine protein-to-creatinine ratio is not indicative of a compromised ability to concentrate urine.

Therefore, this statement is incorrect. Increasing proteinuria, as quantified by the urine protein-to-creatinine ratio, is a strong and independent predictor of chronic kidney disease (CKD) progression, regardless of the underlying etiology. Therapies that significantly reduce proteinuria retard the progression of CKD. Therefore, in this setting of CKD secondary to NPHP-like kidney disease, as evidenced by overall biochemical and histopathological findings, this paradoxical result raises questions about the methodology used to assay urine spot creatinine and protein. The authors should critically examine the methodology used to calculate the spot urine protein-to-creatinine ratio in this animal study and determine whether a true difference in proteinuria exists.

Please see Fig. 3D-F. As requested, we reevaluated protein concentration normalized to creatinine levels using the BCA assay to quantify urine protein levels divided by urine creatinine levels obtained by mass spectrometry from the same urine sample collected in metabolic cages. While previous measurements of total protein and creatinine levels were obtained from the same animals, urine sampling time and collection method were different for these two parameters. Our revised data now show no change in the protein-to-creatinine ratio between wild type and mutant animals. These results are consistent with what is seen in NPHP patients, where polyuria and polydipsia are observed early on, while proteinuria is infrequent and mainly occurs in advanced stages. We have revised the text. Please see lines 17-21, page 8.

3. The acidic urine pH (Figure 3M-O) and low serum tCO₂ levels in aged mice (Appendix Figure S6) further suggested symptoms of acidosis resulting from altered renal handling of electrolytes and solutes. COMMENT: Suggest to remove word “symptoms”. These are findings and not “symptoms”. Please correct the statement after removing the word symptom.
We have revised the text.

26th Mar 2025

Dear Prof. Tsiokas,

I am pleased to inform you that your manuscript is accepted for publication and is now being sent to our publisher to be included in the next available issue of EMBO Molecular Medicine.

If you have any questions, please do not hesitate to contact the Editorial Office.

Thank you for your contribution to EMBO Molecular Medicine!

With kind regards,

Lise Roth
